# Phosphoproteomic mapping reveals distinct signaling actions and activation of muscle protein synthesis by Isthmin-1

Meng Zhao[1,2,3], Niels Banhos Danneskiold-Samsøe[1], Livia Ulicna[1], Quennie Nguyen[1], Laetitia Voilquin[1,2,3], David E Lee[4,5], James P White[4,5,6], Zewen Jiang[1,7,8], Nickeisha Cuthbert[1], Shrika Paramasivam[1], Ewa Bielczyk-Maczynska[2,3,9], Capucine Van Rechem[1], Katrin J Svensson[1,2,3]*

[1]Department of Pathology, Stanford University School of Medicine, Stanford, United States; [2]Stanford Diabetes Research Center, Stanford University School of Medicine, Stanford, United States; [3]Stanford Cardiovascular Institute, Stanford University School of Medicine, Stanford, United States; [4]Duke Molecular Physiology Institute, Duke University School of Medicine, Durham, United States; [5]Department of Medicine, Duke University School of Medicine, Durham, United States; [6]Duke Center for the Study of Aging and Human Development, Duke University School of Medicine, Durham, United States; [7]Department of Laboratory Medicine, University of California, San Francisco, San Francisco, United States; [8]Diabetes Center, University of California, San Francisco, San Francisco, United States; [9]Division of Cardiovascular Medicine, Department of Medicine, Stanford University School of Medicine, Stanford, United States

*For correspondence: katrinjs@stanford.edu

**Abstract** The secreted protein isthmin-1 (Ism1) mitigates diabetes by increasing adipocyte and skeletal muscle glucose uptake by activating the PI3K-Akt pathway. However, while both Ism1 and insulin converge on these common targets, Ism1 has distinct cellular actions suggesting divergence in downstream intracellular signaling pathways. To understand the biological complexity of Ism1 signaling, we performed phosphoproteomic analysis after acute exposure, revealing overlapping and distinct pathways of Ism1 and insulin. We identify a 53% overlap between Ism1 and insulin signaling and Ism1-mediated phosphoproteome-wide alterations in ~450 proteins that are not shared with insulin. Interestingly, we find several unknown phosphorylation sites on proteins related to protein translation, mTOR pathway, and, unexpectedly, muscle function in the Ism1 signaling network. Physiologically, *Ism1* ablation in mice results in altered proteostasis, including lower muscle protein levels under fed and fasted conditions, reduced amino acid incorporation into proteins, and reduced phosphorylation of the key protein synthesis effectors Akt and downstream mTORC1 targets. As metabolic disorders such as diabetes are associated with accelerated loss of skeletal muscle protein content, these studies define a non-canonical mechanism by which this antidiabetic circulating protein controls muscle biology.

## Editor's evaluation

This article will be of interest to those who study integrated physiology by which muscle size, strength, and metabolism are regulated. Effects of the protein Ism1, which is released by adipocytes and immune cells, on the phosphoproteome were compared and contrasted to those of insulin revealing overlapping though distinct signaling pathways. Ism1 was also shown to determine skeletal

muscle size and strength. These data describe a new humoral linkage between fat and skeletal muscle that should have broad implications.

## Introduction

Hormonal signaling through protein phosphorylation is one of the most important post-translational modifications allowing for rapid changes in cellular metabolic states (*Humphrey et al., 2015*). Metabolic stressors such as diabetes or fasting can lead to pronounced physiological and cellular adaptations in protein regulation (*Powers et al., 2009*). Consequently, pathological conditions can result in muscle atrophy, a net loss of muscle mass, which is highly associated with morbidity (*Cohen et al., 2015*; *Jackman and Kandarian, 2004*). Muscle strength is reduced in individuals with insulin resistance and type 2 diabetes (*Andersen et al., 2004*; *Park et al., 2007*), and muscle weakness is even a diagnostic predictor of diabetes (*Peterson et al., 2016*). The mechanisms underlying this association have remained elusive, but it is plausible that factors derived from adipose tissue can contribute to muscle proteostasis. Identifying such molecular triggers of muscle metabolism could facilitate our efforts to develop pharmacological agents that improve muscle function and systemic metabolic health.

Skeletal muscle is the most abundant tissue in humans, representing up to half of the total mass in normal-weight individuals (*Janssen et al., 2000*). As a major organ for glycogen storage and insulin-mediated glucose uptake, skeletal muscle controls whole-body energy expenditure and nutrient homeostasis (*Deshmukh, 2016*; *Shulman et al., 1990*). Importantly, skeletal muscle also acts as a protein reservoir that is highly responsive to anabolic or catabolic hormonal stimulation, including growth hormone (GH) and insulin-like growth factor-1 (IGF-1), both of which stimulate muscle fiber size (hypertrophy) (*Moro et al., 2016*; *Velloso, 2008*). Muscle mass is balanced by pathways controlling protein synthesis and protein degradation. The most well-described anabolic signaling pathway that promotes protein synthesis requires Akt/mTORC1 signaling, which robustly induces muscle hypertrophy upon stimulation by growth factors or amino acids (*Bodine et al., 2001*; *Glass, 2011*; *Lai et al., 2004*). In both flies (*Scanga et al., 2000*) and mammals (*Edinger and Thompson, 2002*), the PI3K-Akt pathway controls cell size by increasing protein synthesis at the level of translation initiation. Akt also inhibits the catabolic function of FoxO family members, which upon phosphorylation are no longer able to enter the nucleus and turn on the transcription of atrophy genes, including the E3 ubiquitin ligases MuRF1 and atrogin-1 (*Bodine et al., 2001*; *Bodine and Baehr, 2014*; *Gomes et al., 2001*; *Lecker et al., 1999*).

We recently reported that the adipocyte-secreted protein isthmin-1 (Ism1) improves glucose tolerance and insulin resistance by phosphorylating Akt$^{S473}$, which mediates increased adipose and skeletal muscle glucose uptake (*Jiang et al., 2021*). *Ism1* adipose expression and circulating levels are elevated in mice and humans with obesity (*Jiang et al., 2021*; *Ruiz-Ojeda et al., 2022*), suggesting that the expression is under nutrient-sensing regulatory control. Intriguingly, Ism1 administration to mice simultaneously prevents hepatic lipid accumulation while increasing protein synthesis in hepatocytes (*Jiang et al., 2021*), demonstrating that Ism1 governs an anabolic pathway that is molecularly and functionally distinct from insulin. In this study, we find that Ism1 induces specific Ism1-regulated phosphoproteome changes enriched for proteins controlling protein translation, mTOR signaling, and muscle function. Furthermore, we show that Ism1 is important to maintain skeletal muscle fiber size under fasting, thereby defining a non-canonical mechanism by which Ism1 controls muscle biology.

## Results

### Phosphoproteomics reveals overlapping and distinct pathways of Ism1 and insulin

The PI3K-Akt pathway is a key pathway of convergence for ligands that activate receptor tyrosine kinases (RTKs), including insulin (*Humphrey et al., 2013*; *Luo et al., 2003*; *Zhao et al., 2020*). We recently identified Ism1 as an adipose-secreted protein that increases glucose uptake into fat and muscle by potently activating PI3K-Akt signaling across a range of mouse and human cell types, but unlike insulin, does not induce de novo lipogenesis (*Jiang et al., 2021*). Therefore, the extent to which the entire Ism1-signaling network overlaps with insulin or whether other signaling nodes

**eLife digest** Cells need energy to survive and carry out their role in the body. They do this by breaking down molecules, like sugar, into substances that can fuel the creation of new compounds, like proteins or lipids. This process, known as metabolism, involves a series of interconnecting chemical reactions which are organized into pathways.

Metabolic pathways contain proteins that catalyze each sequential reaction. Hormones can change the activity of these proteins by adding a chemical group called a phosphate. This reversible modification can majorly impact the metabolism of cells, resulting in changes to the body's tissues. The hormone insulin, for instance, alters a well-known metabolic pathway that triggers skeletal muscle cells to produce more proteins, leading to stronger and larger muscles.

In 2021, a group of scientists discovered a molecule made by fat cells, called Isthmin-1, also activates components in this pathway. Similar to insulin, Isthmin-1 encourages muscle and fat cells to take up sugar. However, it also prevents the liver from accumulating excess fat, suggesting Isthmin-1 may trigger a different cascade of molecules to insulin.

To investigate this possibility, Zhao et al. – including some of the researchers involved in the 2021 study – exposed cells grown in the laboratory to Isthmin-1 or insulin and looked for phosphates on all their proteins. This revealed that only 53% of the proteins Isthmin-1 modifies are also altered by insulin. Of the proteins unique to Isthmin-1, several had known roles in making and maintaining proteins in muscle cells.

To understand more about the role of this newly discovered pathway, Zhao et al. genetically engineered mice to lack the gene that codes for Isthmin-1. This decreased the size and strength of the mice's muscle fibers and reduced the signals that normally lead to skeletal muscle growth.

These findings suggest that Isthmin-1 regulates skeletal muscle size via a metabolic pathway that is slightly different to the one activated by insulin. Many metabolic disorders are associated with muscle loss, like diabetes, and this newly discovered network of proteins could further our understanding of how to prevent and treat these diseases.

are involved remains to be determined. Therefore, to increase our understanding of the signaling divergence and obtain an unbiased, more complete view of the Ism1-induced signaling network, we performed phosphoproteomics in the Ism1 and insulin-responsive 3T3-F442A cells. To characterize the Ism1-induced phospho-signaling profile after acute treatment in cells, we used phosphopeptide enrichment with TiO$_2$ followed by LC-MS/MS using Orbitrap Elite (*Yue et al., 2015*; *Zhou et al., 2008*; *Figure 1A*). Cells were starved overnight, followed by a 5 min treatment with 100 nM recombinant Ism1, or 100 nM insulin. As a negative control, bovine serum albumin (BSA), a secreted protein in the same size range as Ism1, was used at 100 nM. Although pAkt $^{S473}$ induction was more pronounced by insulin, we observed robust activation of pAkt $^{S473}$ 5 min post-treatment with Ism1, and therefore selected this time point for our analysis (*Figure 1B*). The proteomic experiments were performed in treatment groups of six biological replicates, after which the pooled cells were divided into two technical replicates for the proteomics analysis. In total, ~7700 raw MS precursor ions (peptides) were acquired, resulting in the identification of unique phosphopeptides on >5000 proteins (*Figure 1— figure supplement 1A and B*). Principal component analysis (PCA) demonstrates high reproducibility between biological replicates and distinct separation of the Ism1- and insulin-treated groups compared with each other and the albumin control (*Figure 1C*). Interestingly, we identify overlapping and distinct Ism1 and insulin-specific phosphoproteome-wide alterations upon acute stimulation, with BSA as control. There is a 53% overlap between Ism1 and insulin signaling (*Figure 1D*). Insulin induces phosphorylation of 654 phosphosites, out of which 347 phosphosites are also phosphorylated or dephosphorylated by Ism1 (*Figure 1D*).

Remarkably, Ism1 causes changes in the phosphorylation status of 445 proteins compared with BSA that are not shared with insulin and not previously described (*Figure 1D*). Groupwise comparisons between treatments show phosphosites selectively regulated by the specific ligands (adj. p-value of <0.05), many of which have not previously been identified (*Figure 1E*). Based on the notion that Ism1 causes alterations in phosphorylation of a subset of proteins, while another subset is shared with insulin, we next used Gene Ontology (GO) analysis to discern cellular signaling pathways associated

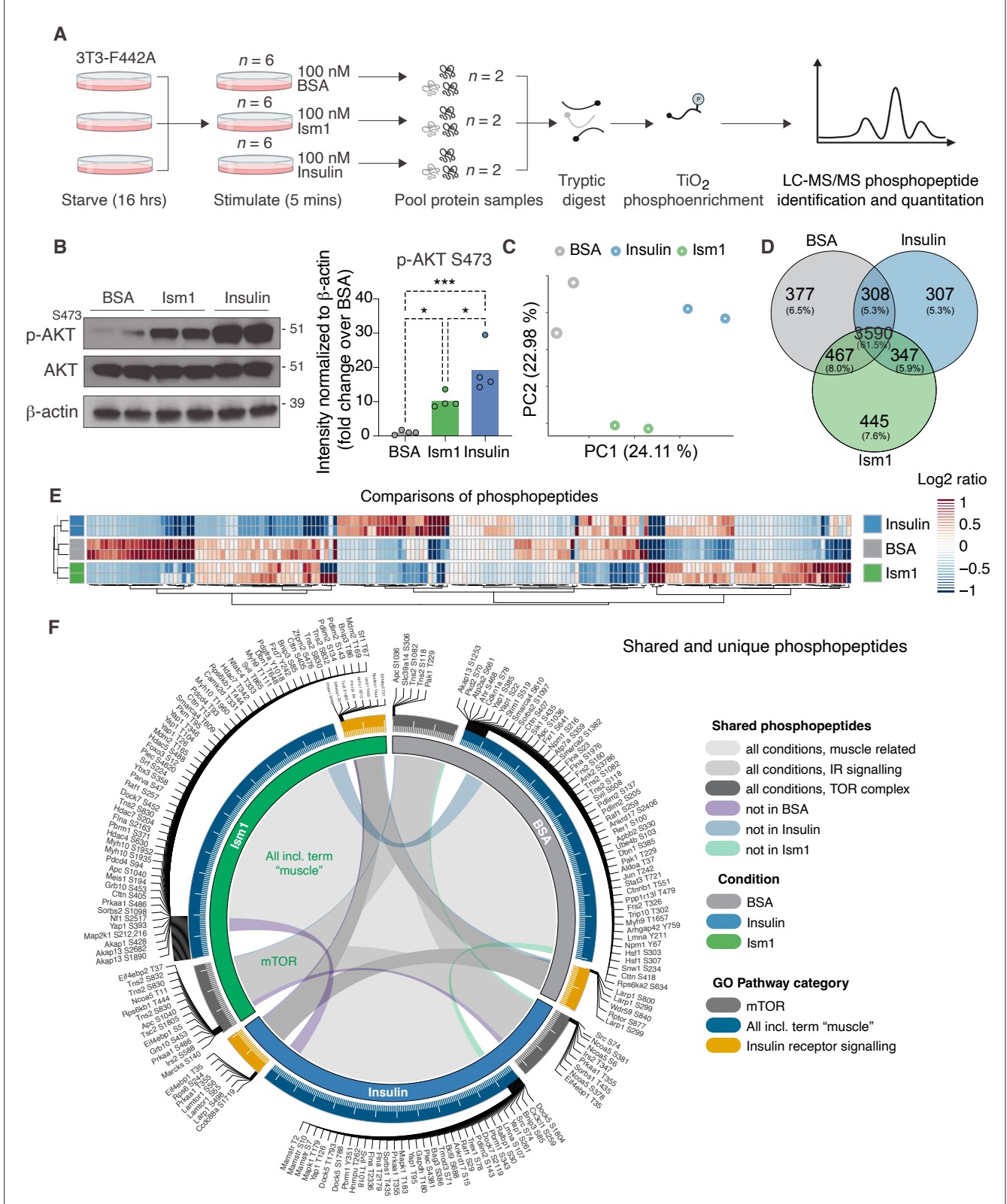

**Figure 1.** Phosphoproteomics reveals overlapping and distinct pathways of Ism1 and insulin. (**A**) Experimental design of the untargeted phosphoproteomics analysis. 3T3-F442A cells were serum-starved for 16 hr and treated with 100 nM recombinant Ism1 or insulin for 5 min (n = 6 biological replicates per group were pooled and then divided into n = 2 technical replicates). Proteins were extracted, trypsin digested, and fractionated. Phosphopeptides were enriched using TiO$_2$ chromatography, and phosphopeptides were analyzed with LC-MS/MS. (**B**) Western blot

*Figure 1 continued on next page*

*Figure 1 continued*

analysis of p-AKT$^{S473}$, total AKT, and β-actin in cells treated with 100 nM bovine serum albumin (BSA), 100 nM Ism1, or 100 nM insulin for 5 min (Western blot n = 2 pooled from n = 6 per treatment group; quantification n = 4 combining three independent assays). p-Values are calculated by one-way ANOVA, *p<0.05, **p<0.01, ***Pp0.001. (**C**) Protein intensity-based principal component analysis (PCA) of the phosphoproteomic dataset. (**D**) Venn diagram of the phosphopeptides detected in both replicates after BSA-, Ism1-, or insulin treatment. (**E**) Heatmap of differentially phosphorylated peptides with 100 lowest significant p-values (adj. p<0.05) from three comparisons displayed as log2 ratio (BSA vs. Ism1, BSA vs. insulin, and insulin vs. Ism1). (**F**) Distribution diagram of shared and unique phosphosites (detected in at least one sample) between treatments for selected Gene Ontology (GO) pathways. Inner links in shades show phosphosites detected in two or more treatment conditions. Gray shows phosphosites detected in all samples; purple shows phosphosites detected in both Ism1 and insulin-treated cells; green shows phosphosites detected in both BSA and insulin-treated cells; blue shows phosphosites detected in both BSA- and Ism1-treated cells. The middle ring displays GO pathway/pathway group with ticks indicating the number of phosphopeptides. Large ticks indicate 50 phosphopeptides, and small ticks indicate 5 phosphopeptides. The outer ring displays gene symbols and the phosphosite exclusively detected in each treatment. See also *Figure 1—figure supplement 1* and *Figure 1—source data 1*, *Figure 1—source data 2*, and *Figure 1—source data 3*.

The online version of this article includes the following source data and figure supplement(s) for figure 1:

**Source data 1.** List of phosphosites significantly different between treatments for *Figure 1E*.

**Source data 2.** List of shared and unique phosphosites between treatments for selected Gene Ontology (GO) pathways for *Figure 1F*.

**Source data 3.** Uncropped Western blot images with relevant bands labeled.

**Figure supplement 1.** Quality controls for the phosphoproteomics analysis.

with Ism1 or insulin. While Ism1 and insulin share the majority of phosphopeptides for detected genes annotated to these GO terms, some phosphorylated residues could only be identified in one condition (*Figure 1F*). Furthermore, we analyzed to which extent the phosphorylation patterns overlapped between the Ism1 and insulin for genes belonging to the GO terms 'GO:0008286, insulin receptor signaling,' 'GO:0038201, TOR complex,' and GO terms including the word 'muscle' (*Figure 1F*). Interestingly, we find that Ism1 exclusively alters the phosphorylation status of proteins involved in the mTOR complex and muscle (*Figure 1F*). These overlapping and distinct signaling nodes may reflect the signaling networks underlying the cell-specific responses.

## Phospho-specific mapping identifies an Ism1-induced signature consistent with protein translation and muscle function

To interrogate the Ism1-specific signal transduction pathway in more detail, we compared the overlapping phosphopeptides clustered by functional GO pathway groups. Expectedly, insulin induces robust phosphorylation of a subset of proteins, including the insulin receptor (IR). This phosphoproteomic mapping shows that Ism1 does not induce the exact same targets in the insulin pathway as insulin (*Figure 2A*), which is entirely consistent with our previous study using phospho-tyrosine antibodies for the IR (*Jiang et al., 2021*). For example, only insulin phosphorylates the InsR at Y1175/Y1163 while no significant phosphorylation is induced by Ism1 (*Figure 2B*). Interestingly, Ism1 induces phosphorylation of some of the same proteins as insulin, including IR substrates Irs1 and Irs2, but with distinct phosphosite patterns. For example, Irs2 was phosphorylated at T347 by insulin but at S588 by Ism1 (*Figure 1F*, *Figure 2—figure supplement 1*). Therefore, Ism1 and insulin activate overlapping but distinct pathways, which may account for some of the phenotypic and cell type-specific functions of Ism1. Similar clustering for the mTOR and muscle pathways (*Figure 2C and D*) revealed several proteins regulated only by insulin or only by Ism1. Insulin induces phosphorylation of ribosomal protein S6 (Rps6), a regulator of protein synthesis downstream of mTOR (*Figure 2E and F*), while Ism1 inhibits Rps6$^{S236}$ and Rps6$^{S240}$ phosphorylation (*Figure 2F*). While the function of these specific phosphosites of Rps6 is understudied, our data suggest a direct regulation of mTOR activity, potentially functionally distinct from insulin. Furthermore, Ism1 induced phosphorylation of several proteins shown to regulate muscle growth and fiber size, such as adenomatous polyposis coli (Apc)$^{S1040}$(*Chen et al., 2020*; *Parisi et al., 2015*; *Figure 2G*), supervillin (Svil)$^{S220}$ (*Hedberg-Oldfors et al., 2020*; *Figure 2H*), and FGF-receptor substrate 2 (Frs2)$^{S327}$ (*Chen and Friesel, 2009*; *Figure 2I*). Frs2 is also a lipid-anchored adapter protein and downstream mediator of signaling of multiple RTKs supporting the existence of a distinct Ism1 receptor (*Zhou et al., 2009*). As protein translation is a critical downstream target of mTOR and functionally controls muscle growth, we next investigated the regulation of ribosomal targets that catalyze protein synthesis. Interestingly, eukaryotic translation initiation factor 3 subunit A (Eif3), a complex-subunit playing a major role in translation initiation (*Ma*

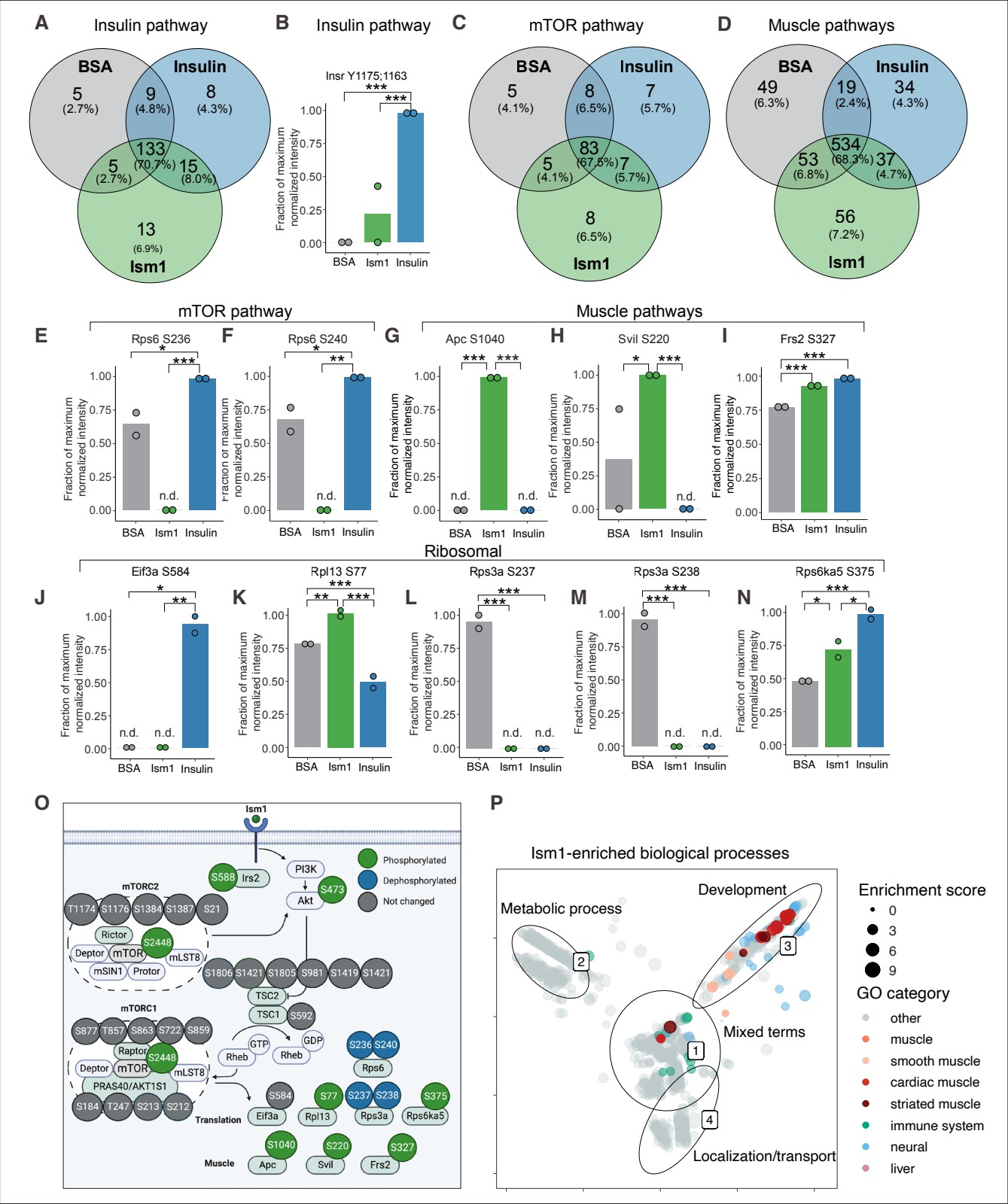

**Figure 2.** Phospho-specific mapping identifies an Ism1-induced signature consistent with protein translation and muscle function. (**A**) Venn diagram of shared and unique phosphosites between treatments for the Gene Ontology (GO) pathways Insulin. (**B**) Abundance of InsR Y1175/1163 in cells treated with bovine serum albumin (BSA), Ism1, or insulin (n = 2). Individual comparisons between conditions across phosphopeptides were performed using empirical Bayes statistics followed by adjustment for multiple testing using false discovery rate, *p<0.05, **p<0.01, ***p<0.001. The minimum

*Figure 2 continued on next page*

*Figure 2 continued*

normalized intensity across the dataset was subtracted from each normalized data point, and phosphorylation was calculated as a fraction of the maximum value of all samples for each phosphopeptide. Bars show mean ± SEM. (C) Venn diagram of shared and unique phosphosites between treatments for the GO pathways Insulin mTOR. (D) Venn diagram of shared and unique phosphosites between treatments for the GO pathways and muscle. (E–N) Abundance of proteins with indicated phosphosite in cells treated with BSA, Ism1, or insulin (n = 2). Individual comparisons between conditions across phosphopeptides were performed using empirical Bayes statistics followed by adjustment for multiple testing using false discovery rate, *$p<0.05$, **$p<0.01$, ***$p<0.001$. The minimum normalized intensity across the dataset was subtracted from each normalized data point, and phosphorylation was calculated as a fraction of the maximum value of all samples for each phosphopeptide. Note that, in case of non-detectable phosphopeptides, significance testing was based on imputed values, n.d., non-detectable. Bars show mean ± SEM. (O) Ism1 signaling network in 3T3-F442A cells. Ism1 ligand stimulation triggers activation of the PI3K/AKT pathway and the mTORC1 pathway, which leads to changes in phosphorylation status of multiple proteins involved in protein translation and muscle function. (P) Pathway analysis of enriched GO pathways in the Ism1 group versus BSA. Clusters are dominated by (1) mixed terms, (2) metabolic process, (3) development, and (4) localization/transport. Plotted GO terms all have p-values <0.01 calculated using the classic Kolmogorov–Smirnov test. See also *Figure 2—figure supplement 1*, *Figure 2—source data 1*, and *Figure 2—source data 2*.

The online version of this article includes the following source data and figure supplement(s) for figure 2:

**Source data 1.** Enriched pathways for proteins with phosphosites significantly different between Ism1 and bovine serum albumin (BSA) clustered by semantic similarity.

**Source data 2.** Raw data related to *Figure 2B,E–N*.

**Figure supplement 1.** Insulin receptor substrate-1 and 2 (Irs1/2) phosphorylation status in response to Ism1 or insulin.

*et al., 2022*), was phosphorylated by insulin but not Ism1 (*Figure 2J*), while 60S ribosomal protein L12 (Rpl13) at S77 (*Figure 2K*) and 40S ribosomal protein S13-1 (Rps13a) at S237 (*Figure 2L*) and Rps3a at S238 (*Figure 2M*) were entirely dephosphorylated by both Ism1 and insulin. Lastly, ribosomal protein S6 kinase alpha-5 (Rps6ka5), a known mTOR substrate (*Chauvin et al., 2014*). was phosphorylated by both Ism1 and insulin at S375 (*Figure 2N*). These data conclude that Ism1 induces phosphorylation of a specific set of proteins involved in mTOR signaling and proteostasis, while other phosphosites are unchanged (*Figure 2O*).

To globally discern possible pathways activated by Ism1 and possible grouping of functional effects, we conducted GO enrichment analysis using biological processes coupled with visualization by semantic similarity. We found associations previously linked to Ism1, including metabolic processes such as glucose and lipid metabolism (*Jiang et al., 2021*; *Ruiz-Ojeda et al., 2022*), nervous system development (*Osório et al., 2014*; *Pera et al., 2002*), and the immune system (*Lam et al., 2022*; *Li et al., 2021*; *Valle-Rios et al., 2014*; *Wu et al., 2021*; *Figure 2P*). Intriguingly, also here, we identify several pathways associated with muscle, skeletal muscle, and cardiac muscle in the Ism1 treatment group (*Figure 2P*). These results show a broad regulation of signatures indicating a role for Ism1 in muscle function.

## Ism1 induces mTOR-dependent protein synthesis in muscle cells

Given the Ism1-induced muscle-signaling signature in 3T3-F442A cells and that the PI3K-Akt pathway is known to promote anabolic programs in muscle cells (*Edinger and Thompson, 2002*), muscle cell differentiation (*Wilson and Rotwein, 2007*), and skeletal muscle hypertrophy (*Bodine et al., 2001*; *Jaiswal et al., 2019*), we next asked whether Ism1 induces anabolic cellular signaling pathways in muscle cells. In differentiated C2C12 myotubes, Ism1 induces phosphorylation of Akt[S473] and ribosomal S6[S235/S236] starting at 5 min and remaining up to 4 hr (*Figure 3A*). The effect of Ism1 on Akt signaling is robust but lower than that of the skeletal muscle hypertrophy hormone insulin-like growth factor-1 (Igf1) (*Figure 3A and B*). Similarly, undifferentiated C2C12 myoblasts are also responsive to Ism1 in a dose-dependent manner (*Figure 3C*). Notably, Ism1 treatment induced a 2.5-fold increase in protein synthesis as determined by [35S]-methionine incorporation into proteins (*Figure 3D*). As expected, Igf1 treatment resulted in a threefold induction in protein synthesis (*Figure 3D*), and the combined Igf1 and Ism1 treatment did not induce protein synthesis further, suggesting that the maximal capacity of protein synthesis has been reached under these conditions. Previous data showed that mTORC1/2 inhibition with rapamycin inhibits Ism1-induced signaling in 3T3-F442A cells, demonstrating that intact mTOR activity is required for the signaling capacity of Ism1 (*Jiang et al., 2021*). Importantly, low-dose rapamycin also inhibits Ism1-induced protein synthesis, establishing that the functional effects are directly linked to the signaling cascade induced by Ism1 (*Figure 3E*). These

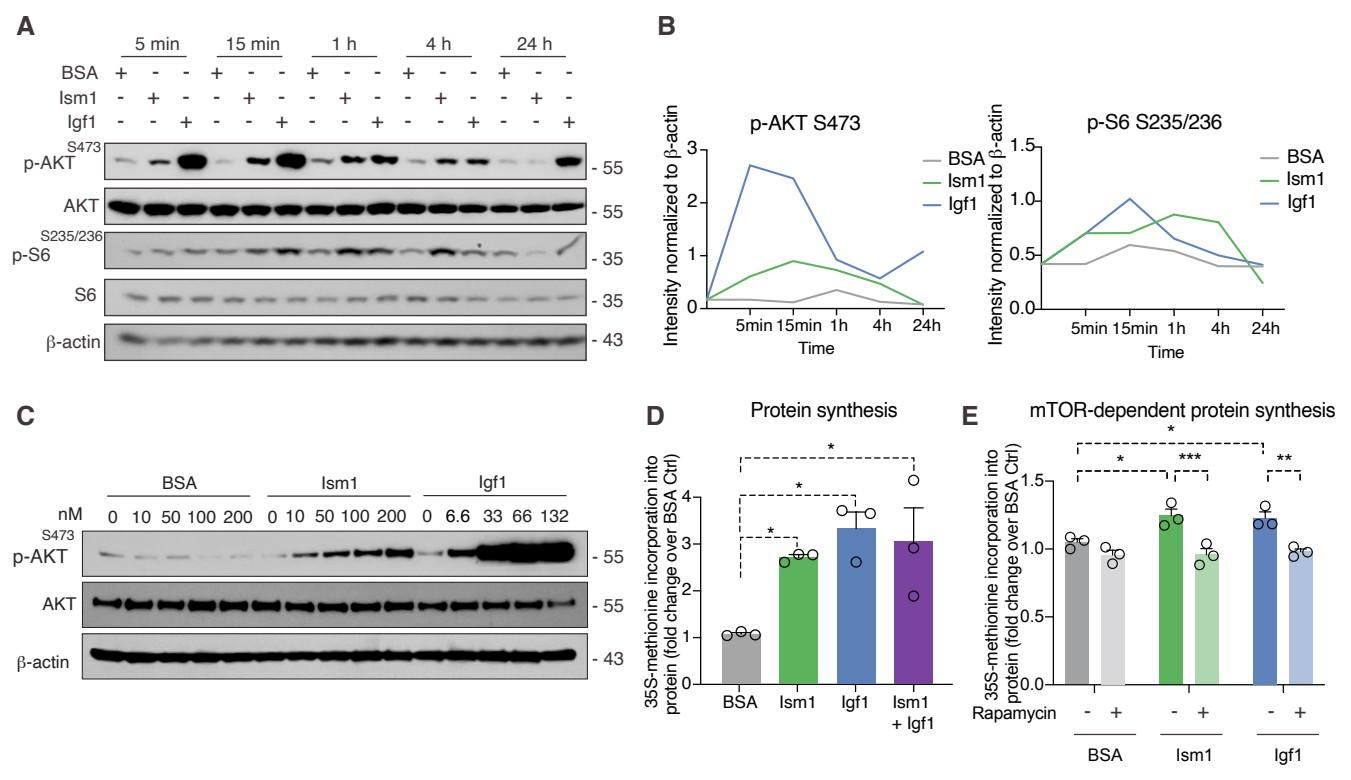

**Figure 3.** Ism1 induces mTOR-dependent protein synthesis in muscle cells. (**A**) Western blot analysis of p-AKT$^{S473}$, total AKT, p-S6$^{S235/}$236, total S6, and β-actin in C2C12 myotubes treated with 100 nM bovine serum albumin (BSA), 100 nM Ism1, or 50 ng/ml Igf1. (**B**) Quantification of protein expression of p-AKT S473/β-actin and p-S6 S235/236/β-actin. (**C**) Western blot analysis of p-AKT$^{S473}$, total AKT, and β-actin in C2C12 myoblasts treated with indicated concentrations of BSA, Ism1, or Igf1 for 5 min. (**D**) Levels of protein synthesis measured by [$^{35}$S]-methionine incorporation in C2C12 myotubes after 48 hr of indicated treatments (n = 3, one-way ANOVA, *p<0.05, **p<0.01, ***p<0.001). (**E**) Levels of protein synthesis measured by [$^{35}$S]-methionine incorporation in C2C12 myotubes with indicated treatments for 2 hr in the presence or absence of 100 nM of the mTOR inhibitor rapamycin (n = 3, two-way ANOVA, *p<0.05, **p<0.01, ***p<0.001). Bar graphs show mean ± SEM. See also *Figure 3—source data 1* and *Figure 3—source data 2*.

The online version of this article includes the following source data for figure 3:

**Source data 1.** Raw data related to *Figure 3B, D, and E*.

**Source data 2.** Uncropped Western blot images with relevant bands labeled.

data align with the inhibitory effects of rapamycin on muscle hypertrophy during anabolic conditions (*Pallafacchina et al., 2002*). In conclusion, Ism1 induces a signaling cascade that requires intact mTOR signaling to induce protein synthesis in muscle cells.

## *Ism1* ablation results in reduced skeletal muscle fiber size and muscle strength

Skeletal muscle atrophy, a reduction in muscle mass, occurs when the protein degradation rate exceeds protein synthesis (*Cohen et al., 2015*; *Jaiswal et al., 2019*; *Sandri et al., 2004*). We previously showed that Ism1 controls glucose uptake into adipose tissue and skeletal muscle in mice (*Jiang et al., 2021*), but the role of Ism1 in skeletal muscle function beyond glucose regulation has not been studied. Ism1 is broadly expressed, including highly in adipose tissue and blood (*Jiang et al., 2021*). By analyzing single-cell RNA sequencing data from murine skeletal muscle (*Baht et al., 2020*), we find that *Ism1* is not expressed in muscle precursors or mature muscle cells (*Figure 4—figure supplement 1*). Given the observed signaling action of Ism1 on muscle cells, this indicated a non-cell-autonomous effect of Ism1 on muscle cells in a physiological setting. Therefore, we next sought to determine whether whole-body ablation of *Ism1* in mice, in which blood levels of Ism1 are completely ablated (*Jiang et al., 2021*), resulted in changes to skeletal muscle mass. To evaluate muscle atrophy, 8-week-old WT and *Ism1*-KO mice were either fasted for 12 hr or fasted followed by a 12 hr re-feeding period (fed state) (*Figure 4A*). As expected, at dissection, all fasted mice had a reduction in body

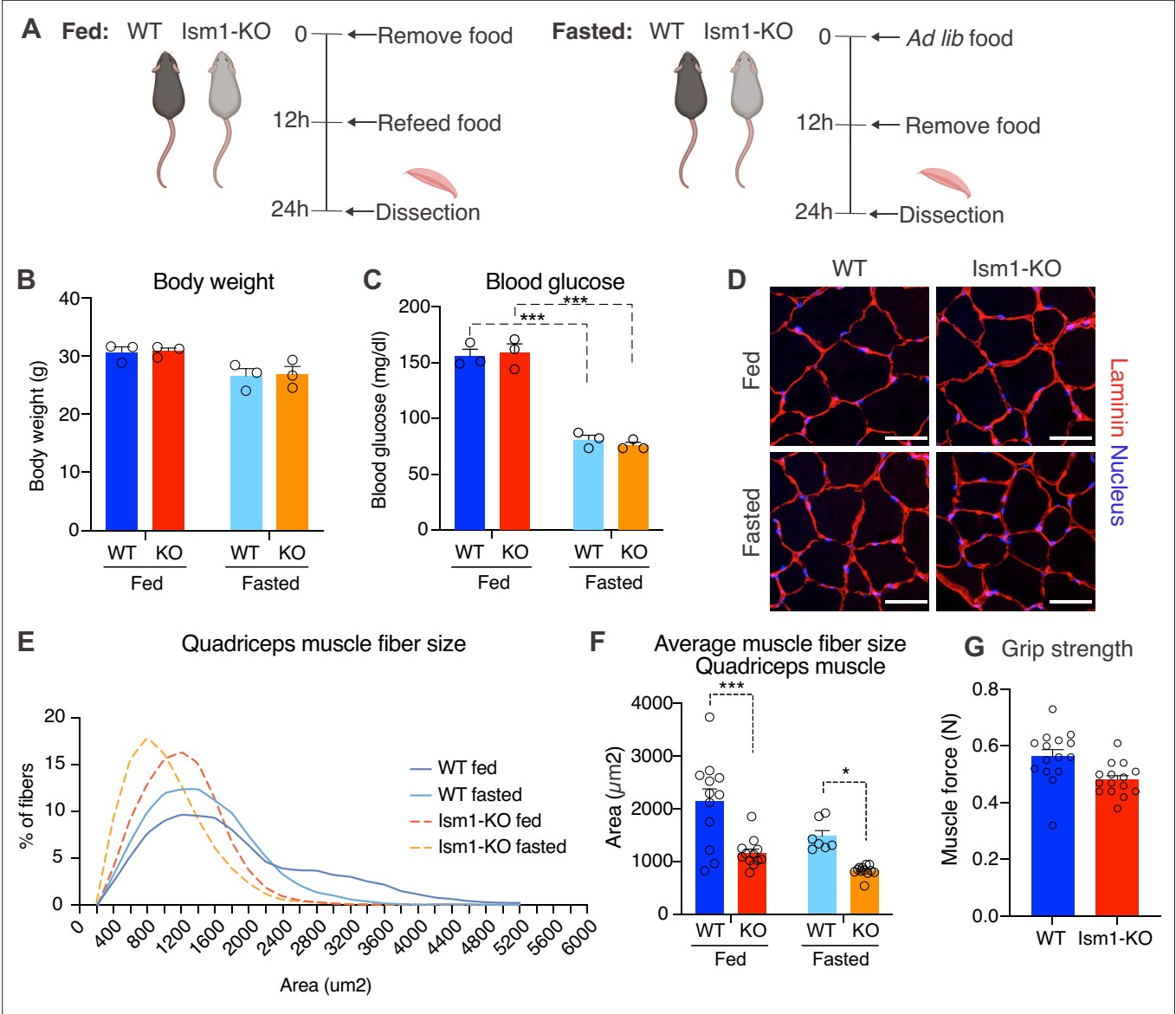

**Figure 4.** *Ism1* ablation results in reduced skeletal muscle fiber size and muscle strength. (**A**) Schematic description of the fasting and feeding protocol. (**B**) Body weights of WT and *Ism1*-KO mice in the fed or fasted groups (WT fed, n = 3; *Ism1*-KO fed, n = 3; WT fasted, n = 3; *Ism1*-KO fasted, n = 3, two-way ANOVA). (**C**) Blood glucose level in fed and fasted mice before dissection (WT fed, n = 3; *Ism1*-KO fed, n = 3; WT fasted, n = 3; *Ism1*-KO fasted, n = 3, two-way ANOVA, *p<0.05, **p<0.01, ***p<0.001). (**D**) Immunolabeling of laminin (red), staining of nucleus (blue) of mouse quadriceps muscles from WT and *Ism1*-KO mice in the fed or fasted groups (scale bars: 10 μm). Images are representative examples of three mice showing similar results. (**E**) Fiber size distribution of mouse quad muscles (WT fed, n = 3; *Ism1*-KO fed, n = 3; WT fasted, n = 3; *Ism1*-KO fasted, n = 3). Mean percentage of myofibers within the indicated range is shown. (**F**) Quantification of average muscle fiber area (WT fed, n = 3; *Ism1*-KO fed, n = 3; WT fasted, n = 3; *Ism1*-KO fasted, n = 3, one muscle tissue per mouse, 2–4 images per muscle tissue,~60–100 myofibers quantified per image; two-way ANOVA, *p<0.05, **p<0.01, ***p<0.001) performed in a blinded fashion by two independent investigators. This experiment was repeated using two independent cohorts of mice. (**G**) Grip strength measured by two-paw muscle force (**N**) on a grid in WT and *Ism1*-KO mice (WT, n = 15; *Ism1*-KO, n = 15). p-Values are calculated by two-tailed Student's *t*-test, *p<0.05, **p<0.01, ***p<0.001. Bar graphs show mean ± SEM. See also *Figure 4—figure supplements 1 and 2* and *Figure 4—source data 1*.

The online version of this article includes the following source data and figure supplement(s) for figure 4:

**Source data 1.** Raw data related to *Figure 4B, C,E–G*.

**Figure supplement 1.** Ism1 acts non-cell-autonomously on muscle cells.

**Figure supplement 2.** *Ism1* ablation does not reduce muscle mass or muscle fiber size in all tissue locations.

weight compared with fed mice (*Figure 4B*). Blood glucose levels were 150 mg/dl in the fed state, and <80 mg/dl in the fasted state (*Figure 4C*), but no differences were seen between genotypes. Under the same conditions, quadriceps (*Figure 4D*), gastrocnemius (*Figure 4—figure supplement 2*), soleus (*Figure 4—figure supplement 2*), as well as tibialis (*Figure 4—figure supplement 2*) muscles were harvested from mice under fed or fasted states and analyzed for muscle fiber size. Muscle tissue morphology was similar between the genotypes, but the muscle fiber size was notably smaller in the *Ism1*-KO quadriceps muscles (*Figure 4D*). Remarkably, fiber size area quantifications showed that the *Ism1*-KO quadriceps muscles demonstrate a robust shift in distribution to smaller muscle fiber size areas under both fed and fasted conditions (*Figure 4E*). On average, loss of *Ism1* significantly reduces the cross-sectional area by 40% in the fed state and >20% in the fasted state (*Figure 4F*). Given that the Ism1 phenotype was more pronounced under the fed state, we next evaluated the effect of *Ism1* ablation on muscle function. While the heart weights (*Figure 4—figure supplement 2*), muscle weights (*Figure 4—figure supplement 2*), femur length (*Figure 4—figure supplement 2*), or total body weights (*Figure 4B*) of *Ism1*-KO mice were not significantly different from WT, the *Ism1*-KO mice had impaired muscle force (*Figure 4G*). In conclusion, these results show that *Ism1* ablation leads to smaller skeletal muscle fiber size and loss of muscle strength.

## Ism1 ablation does not impair movement or mitochondrial biogenesis, or normal muscle development

Since *Ism1* ablation causes severe myofiber atrophy, we next investigated whether loss of *Ism1* is associated with a reduction in movement or muscle mitochondrial complex-dependent mitochondrial bioenergetics, as has been demonstrated for IR and IGF-1R (*Bhardwaj et al., 2021*). Ambulatory activity (*Figure 5A*), respiratory exchange ratio (RER) (*Figure 5B*), energy expenditure as measured by $VO_2$ consumption (*Figure 5C*), or food intake (*Figure 5D*) are indistinguishable between WT and *Ism1*-KO mice when measured over a 48 hr period. Furthermore, protein levels of the mitochondrial complexes under fed and fasted conditions show no significant differences in any of the mitochondrial OXPHOS complexes I, II, III, IV, or V (*Figure 5E and F*). These data are consistent with the notion that the expression of the canonical regulator of mitochondrial biogenesis, *Ppargc1a* (*Pgc1α*) (*Lin et al., 2002*; *Wu et al., 1999*), is unchanged in quadriceps muscle tissues from the *Ism1*-KO mice (*Figure 5G*). General muscle markers, including the myosin heavy chain proteins *Myh1*, *Myh2*, *Myh4*, and *Myh7*, were not altered under fed conditions, and only *Myh4* was reduced in the *Ism1*-KO mice under fasted conditions, suggesting that muscle fiber type is largely unaffected by Ism1 ablation (*Figure 5H*). Taken together, these data suggest that Ism1 regulates muscle fiber size and muscle strength without affecting the expression of mitochondrial complexes or whole-body energy expenditure.

## *Ism1*-KO mice have defective skeletal muscle Akt and mTOR signaling and protein synthesis

To understand the underlying mechanism by which *Ism1*-KO mice develop smaller muscle fiber size, we next asked whether Ism1 directly controls protein content in mice. Quadriceps muscles from 8-week-old mice were harvested and analyzed for total protein content, demonstrating a significant reduction in total protein content in the *Ism1*-KO mice compared with WT mice (*Figure 6A*). Because muscle proteostasis is balanced by protein synthesis and degradation (*Glass, 2011*; *Jackman and Kandarian, 2004*), we next tested the hypothesis that Ism1 is required for efficient protein synthesis in mice by measuring in vivo [$^{35}$S]-methionine incorporation into proteins isolated from quadriceps muscles (*Figure 6B*). WT and *Ism1*-KO mice were i.p. administered with [$^{35}$S]-methionine for 2 hr, followed by protein precipitation, which demonstrates that loss of Ism1 results in a significant reduction in muscle protein synthesis in mice (*Figure 6C*). Furthermore, under either fed or fasted conditions, muscles isolated from *Ism1*-KO mice have significantly increased transcript levels of atrophy genes FoxO1 (*Figure 6D*), the FoxO target genes *Cdkn1b*, *Eif4ebp1*, and *Ctsl* (*Figure 6E*), and ubiquitin ligase *Fbxo30* compared with WT mice (*Figure 6—figure supplement 1*), and increased protein levels of the FoxO target p27 (*Cdkn1b*) (*Zhang et al., 2011*; *Figure 6F*). These data suggest that Ism1 controls protein synthesis, and its loss is associated with elevated protein degradation gene expression. Given that Ism1 activates the Akt pathway in myocytes in vitro, and that loss of muscle-specific Akt1 and Akt2 leads to muscle atrophy (*Jaiswal et al., 2019*), we next hypothesized that the mechanism behind the smaller muscle fiber size in the *Ism1*-KO mice is due to reduced Akt pathway activity.

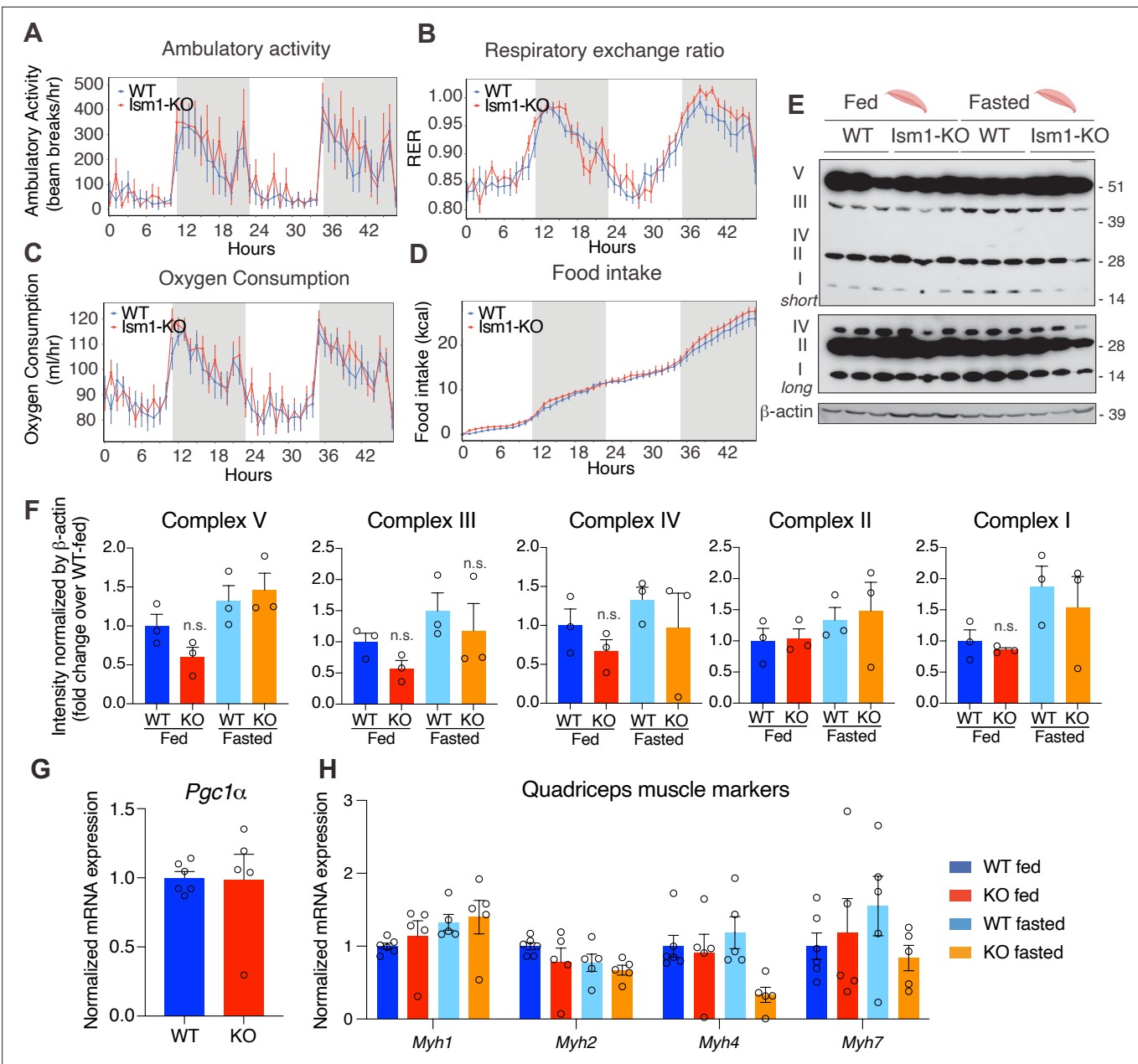

**Figure 5.** *Ism1* ablation does not impair movement, mitochondrial biogenesis, or normal muscle development. (**A**) Ambulatory activity (WT, n = 4; *Ism1*-KO, n = 4, ANOVA, *p<0.05). (**B**) Respiratory exchange ratio (RER) (WT, n = 4; *Ism1*-KO, n = 4, ANOVA, *p<0.05). (**C**) Oxygen consumption (WT, n = 4; *Ism1*-KO, n = 4, ANCOVA, *p<0.05). (**D**) Food intake (WT, n = 4; *Ism1*-KO, n = 4, ANCOVA, *p<0.05) in WT and Ism1-KO mice. Mice were habituated for 24 hr followed by 48 hr recordings of metabolic parameters. (**E**) Levels of mitochondrial oxidative phosphorylation proteins in ETC complexes (OXPHOS) from quadriceps muscles (WT fed, n = 3; *Ism1*-KO fed, n = 3; WT fasted, n = 3; *Ism1*-KO fasted, n = 3) analyzed by Western blot. (**F**) Quantification of OXPHOS complexes (two-way ANOVA, *p<0.05, **p<0.01, ***p<0.001). (**G**) Relative gene expression analysis of *Pgc1α* in quadriceps muscle from WT (n = 6) or *Ism1*-KO (n = 5) mice (two-tailed Student's *t*-test, *p<0.05, **p<0.01, ***p<0.001). (**H**) Relative gene expression analysis of *Myh1, Myh2, Myh4*, and *Myh7* in quadriceps muscle from WT fed (n = 6) or *Ism1*-KO fed (n = 5) vs. WT fasted (n = 5) or *Ism1*-KO fasted (n = 5) mice (two-way ANOVA, *p<0.05, **p<0.01, ***p<0.001). Bar graphs show mean ± SEM. See also *Figure 5—source data 1* and *Figure 5—source data 2*.

The online version of this article includes the following source data for figure 5:

**Source data 1.** Raw data related to *Figure 5A–D,F–H*.

**Source data 2.** Uncropped Western blot images with relevant bands labeled.

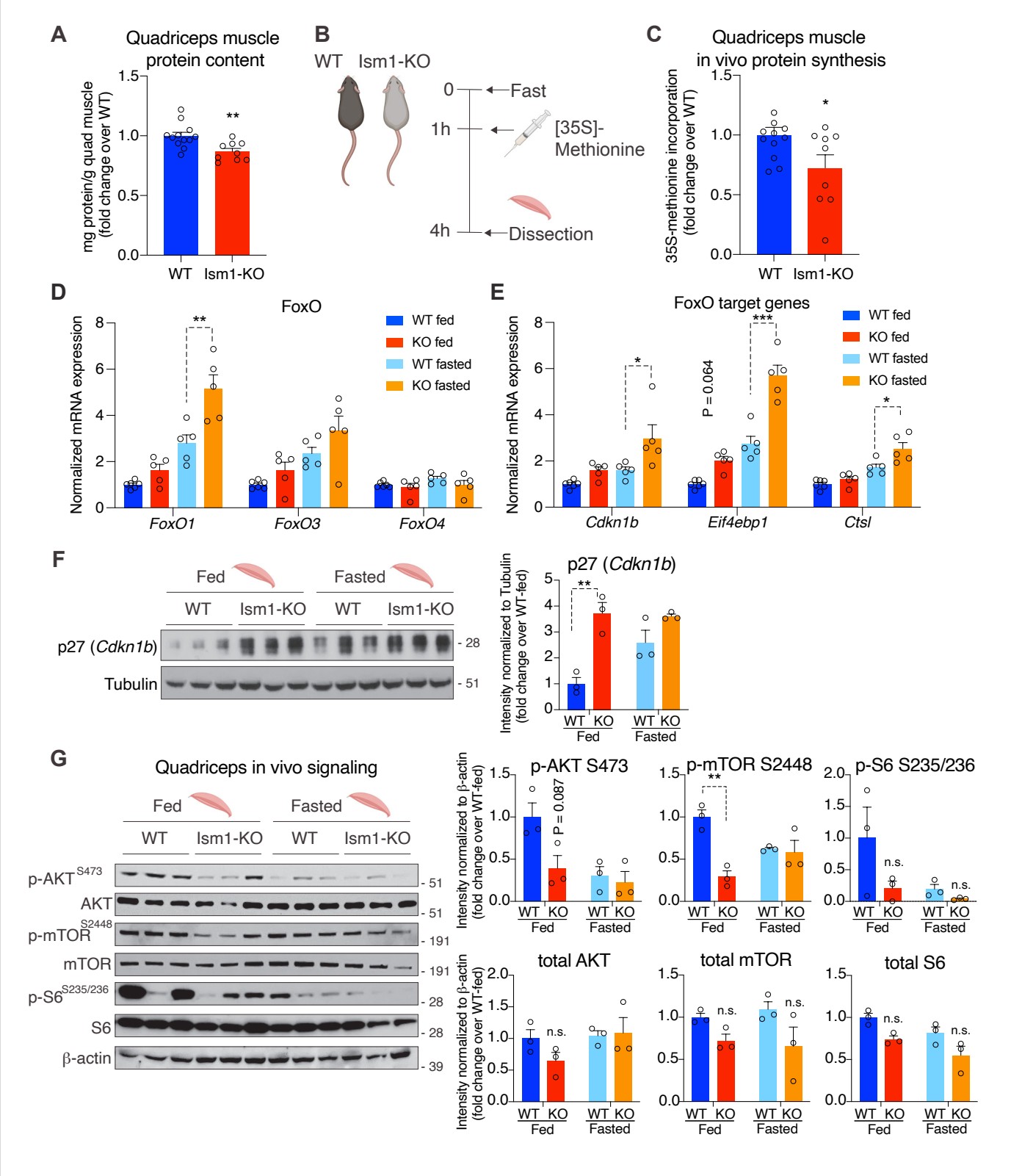

**Figure 6.** *Ism1*-KO mice have defective skeletal muscle protein synthesis and AKT-mTOR signaling. (**A**) Total protein content measured in WT and *Ism1*-KO quadriceps muscle expressed as mg protein/wet tissue weight in grams. (WT, n = 12; *Ism1*-KO, n = 9, two-tailed Student's *t*-test, *p<0.05, **p<0.01, ***p<0.001). (**B**) Schematic description of the in vivo [35S]-methionine incorporation protocol. (**C**) In vivo protein synthesis measured by [35S]-methionine incorporation in WT and *Ism1*-KO mice (WT, n = 12; *Ism1*-KO, n = 9, two-tailed Student's *t*-test, *p<0.05, **p<0.01, ***p<0.001). Relative gene

*Figure 6 continued*

expression analysis of (**D**) *FoxO* and (**E**) *FoxO* target genes *Cdkn1b, Eif4ebp1,* and *Ctsl* in quadriceps muscle from WT fed (n = 6) or *Ism1*-KO fed (n = 5) vs. WT fasted (n = 5) or *Ism1*-KO fasted (n = 5) mice (two-way ANOVA, *p<0.05, **p<0.01, ***p<0.001). (**F**) Western blot analysis and quantification of the levels of p27 and tubulin in the cytosolic fraction of quadriceps muscles of WT and *Ism1*-KO mice under fed and fasted conditions (WT fed, n = 3; *Ism1*-KO fed, n = 3, WT fasted, n = 3; *Ism1*-KO fasted, n = 3, two-way ANOVA, *p<0.05, **p<0.01, ***p<0.001). (**G**) Western blot analysis and quantification of the levels of pAKT$^{S473}$, total AKT, p-mTOR$^{S2448}$, total mTOR, pS6$^{S235/236}$, total S6, and β-actin in quadriceps muscles of WT and *Ism1*-KO mice under fed and fasted conditions (WT fed, n = 3; *Ism1*-KO fed, n = 3, WT fasted, n = 3; *Ism1*-KO fasted, n = 3, two-way ANOVA, *p<0.05, **p<0.01, ***p<0.001). Bar graphs show mean ± SEM. See also *Figure 6—figure supplement 1*, *Figure 6—source data 1*, and *Figure 6—source data 2*.

The online version of this article includes the following source data and figure supplement(s) for figure 6:

**Source data 1.** Raw data related to *Figure 6A,C–G*.

**Source data 2.** Uncropped Western blot images with relevant bands labeled.

**Figure supplement 1.** Ubiquitin expression and insulin receptor substrate-1 (Irs1/2) phosphorylation status in quadriceps muscles of WT and Ism1-KO mice.

Indeed, we found that phosphorylation of Akt$^{S473}$, mTOR$^{2448}$, and ribosomal S6$^{S235/S236}$ are markedly decreased in *Ism1*-KO muscle compared with WT mice under both fed and fast conditions, providing an explanation for the lower protein synthesis rate (*Figure 6G*). *Ism1*-KO mice are indistinguishable from WT mice in the phosphorylation of IRS1 at S307 (*Figure 6—figure supplement 1*), a canonical IRS phosphorylation site (*Rui et al., 2001*). Taken together, these results demonstrate that Ism1 is an anabolic regulator of protein synthesis and muscle strength by maintaining high Akt activity and protein synthesis in skeletal muscle.

## Discussion

Akt has an established role in enhancing muscle hypertrophy and function (*Bodine et al., 2001*; *Glass, 2011*; *Jaiswal et al., 2022*; *Jaiswal et al., 2019*; *Lai et al., 2004*; *Mammucari et al., 2007*; *Wilson and Rotwein, 2007*). However, there is still a need to identify other hormonal and physiological insulin/IGF-1 independent activators of Akt to avoid associated side effects such as hypoglycemia when used therapeutically. We previously identified Ism1 as a secreted protein that activates Akt in multiple cell types, including human skeletal muscle cells. Using radiolabeled glucose, we also showed that Ism1 increases glucose transport into both adipose tissue and skeletal muscle (*Jiang et al., 2021*). Several lines of evidence from this study suggest that Ism1 also has an important anabolic role in promoting skeletal muscle growth. Most importantly, Ism1 acts directly on muscle cells to induce protein synthesis. Conversely, *Ism1* ablation leads to lower skeletal muscle fiber size and atrophy associated with lower Akt signaling and protein synthesis. *Ism1*-KO mice have elevated levels of FoxO1 target genes consistent with increased protein degradation. Therefore, Ism1 appears to be an important protein regulating metabolism during fed and fasted conditions.

Ism1-induced signaling is not simply an activation of the identical insulin/IGF-1-induced PI3K- Akt pathway. Across multiple cell types, Ism1 and insulin share the pAkt and pS6 network but are seemingly segregated by the more robust activation of pAkt$^{S473}$ seen with insulin, whereas Ism1 induces only a subset of shared Akt-induced insulin targets. Interestingly, while Ism1 does not directly phosphorylate the IR/IGFRs, Ism1 does induce phosphorylation of the IR substrate proteins (Irs), a feature that is shared with other endocrine hormones. This phosphospecific regulation by distinct hormones can result in a diverse range of functional outcomes (*Yenush and White, 1997*). For example, Irs1 phosphorylation at S302, S307, S522, and S636/639 have been linked to insulin resistance (*Um et al., 2004*), but not all hormones that phosphorylate those Irs sites induce insulin resistance, including FGF21 (*Minard et al., 2016*). These overlapping but distinct pathways may account for the Ism1's cell type-specific functional outcomes and downstream transcriptional effects unique to Ism1. Additionally, the phosphoproteomic mapping shows a distinct muscle signature specifically induced by Ism1 and not by insulin. Among Ism1 downstream targets, we find phosphorylation of proteins related to the mTOR pathway, ribosomal and muscle function, targets not previously identified downstream of Ism1.

In this work, we observe a 60% decrease in pAkt$^{S473}$ levels in the muscles of *Ism1*-KO mice, which leads to an ~10% reduction in muscle protein content. Our data is consistent with previous reports using muscle-specific *Akt*-KO mice that demonstrate a 40% reduction in protein synthesis upon

complete ablation of Akt (*Jaiswal et al., 2022*). Furthermore, reduced Akt is associated with skeletal muscle insulin resistance, in line with our previous observation that *Ism1*-KO mice fed a high-fat diet are more insulin resistant (*Jiang et al., 2021*). To what extent the insulin resistance in the *Ism1*-KO mice contributes to muscle function in obesity or during aging is an intriguing question that remains to be answered in future work. Moreover, increasing protein synthesis while suppressing protein degradation is important in a physiological setting, based on the findings that only a combination of FoxO1 inhibition and mTORC1 activation can restore Akt-mediated muscle loss (*Jaiswal et al., 2022*). This suggests that hormonal regulators of Akt activity could have important biological functions because of Akt's dual role in regulating mTORC1 and the FoxO genes. Here, we used genetic and pharmacological approaches to determine the role of Ism1 in regulating protein synthesis, but it remains to be explored whether pharmacological Ism1 administration prevents protein degradation in the skeletal muscle. Additionally, upon muscle hypertrophy, activated muscle precursors or satellite cells provide an additional mechanism for muscle expansion (*Hawke and Garry, 2001*). A limitation of this study is that we did not investigate whether Ism1 affects muscle regeneration. As IGF-1 induces the growth and differentiation of satellite cells (*Musarò et al., 1999*), and HGF induces activation of cell cycle $G_{Alert}$ (*Rodgers et al., 2017*), it will be interesting to determine whether Ism1 also activates skeletal muscle stem cells. Identifying all physiological mechanisms that distinguish Ism1 from IGF-1 will aid in understanding whether other yet unidentified pathways distinct from Akt and FoxO regulate proteostasis.

The physiological function of Ism1 in regulating muscle growth is expected given that Ism1 stimulates Akt, but it was somewhat unexpected that the lower muscle protein content and fiber size did not lead to a significant loss of muscle mass. Femur length is a basic biometric parameter used to assess longitudinal growth. Femur length and total body weights did not differ between WT and *Ism1*-KO mice, suggesting that the decrease in muscle fiber size is likely not due to growth reduction. It remains to be determined what replaces protein mass in the muscle of *Ism1*-KO mice, including water, glycogen stores, or lipids, which could be regulated by Ism1. Further studies using larger sample sizes are needed to determine whether the effects of Ism1 on muscle proteostasis are specific to fiber types and locations. Notably, *Ism1* itself is not expressed in muscle cells but only in other cells, including adipocytes and immune cells. Circulating levels of human Ism1 using immunoreactive ELISAs have been reported to be in the range of 1–50 ng/ml (*Jiang et al., 2021*; *Ruiz-Ojeda et al., 2022*), consistent with mass spectrometry peptide analysis that estimates a circulating Ism1 concentration of 17 ng/ml (*Uhlén et al., 2019*). The necessary circulating and local Ism1 levels to maintain muscle fiber size or prevent fasting-induced muscle loss remain to be determined. It is plausible that Ism1 levels are altered in response to fasting and feeding, insulin resistance, or aging – conditions associated with temporary or chronic muscle loss (*Laurens et al., 2021*; *Perry et al., 2016*; *Roubenoff and Castaneda, 2001*). As this study was conducted in male mice only, it is unknown whether Ism1 deficiency results in any muscle phenotype in female mice. Further studies are required to test whether Ism1 regulates muscle proteostasis in a sex-specific manner. A limitation of this study is that fasting but not more severe atrophy conditions were studied. Long-term studies ought to explore whether pharmacological administration of Ism1 systemically or locally into the muscle is sufficient to induce muscle hypertrophy or prevent muscle loss in more severe models of muscle loss. Nevertheless, as aging impairs skeletal muscle protein synthesis and leads to muscle weakness and atrophy, this work has implications for multiple conditions, including diabetes- and age-induced muscle atrophy.

## Materials and methods
### Animal studies

Animal experiments were performed per procedures approved by the Institutional Animal Care and Use Committee of the Stanford Animal Care and Use Committee (APLAC) protocol number #32982. C57BL/6J mice were purchased from the Jackson Laboratory (#000664). The *Ism1*-KO mice (C57BL/6J-Ism1[em1Kajs]/J strain #036776, JAX) were generated using the *Ism1* floxed allele as described previously (*Jiang et al., 2021*). Unless otherwise stated, mice were maintained on a chow diet (Envigo Teklad Global 18% Protein Rodent Diet 2018) and housed in a temperature-controlled (20–22°C) room on a 12 hr light/dark cycle. All experiments were performed with age-matched male mice housed in groups of five unless stated otherwise.

## Sample preparation for phosphoproteomics analysis

The phosphoproteomics analysis was performed using 3T3-F442A cells. The 3T3-F442A cell line is mycoplasma negative and has been authenticated with STR. Cells were cultured in DMEM/F12 medium with 10% FBS/1% pen/strep until 80–100% confluent. Following a 16 hr starvation in serum-free DMEM/F12 medium, cells were treated with PBS, 100 nM Ism1 or 100 nM insulin for 5 min (N = 6 per treatment group, 10M cells/treatment). Following treatment, the medium was aspirated, and cells were washed three times with ice-cold PBS while kept on ice. Then, 1 ml ice-cold PBS supplemented with cOmplete Mini Protease Inhibitors (#4693124001, Sigma-Aldrich, St. Louis, MO) and phosSTOP phosphatase inhibitors (#4906845001, Sigma-Aldrich) were added to the cells that were immediately scraped down, pelleted at 14,000 × $g$ for 10 min followed by snap freezing. Cell pellets were kept at –80°C until analysis. Aliquots of the samples were used to verify AKT$^{S473}$ signaling induction by Ism1 and insulin by Western blot. For each treatment group, six biological replicates were pooled into two technical replicates for the phosphoproteomics analysis. The phosphoproteomics analysis was performed at Northwestern University Proteomics core as described previously (*Dephoure et al., 2013*). Briefly, protein extracts were alkylated with iodoacetamide and digested with trypsin at a ratio of 1:50 (w/w) trypsin to protein. Tryptic phosphopeptides were enriched by TiO$_2$-immobilized metal affinity chromatography, washed with 70% (v/v) EtOH, and equilibrated in 1% NH$_4$OH and three times in 1 M glycolic acid in 80% (v/v) acetonitrile, 5% (v/v) trifluoroacetic acid. Peptides were eluted and dried in a SpeedVac concentrator. Peptides were resuspended in 3% (v/v) acetonitrile, 0.1% trifluoro-acetic acid, and twice in 0.1% trifluoroacetic acid. Dried peptides were dissolved in LC-MS/MS solvent (3% acetonitrile and 0.1% fluoroacetic acid) prior to LC-MS/MS analysis using a dual-pressure linear ion trap (Velos Pro) with a high-field Orbitrap mass analyzer as described previously (*Yue et al., 2015*; *Zhou et al., 2008*) using MaxQuant to analyze raw spectrometry data (*Tyanova et al., 2016*).

## Phosphoproteomic analysis

Data acquired on the Orbitrap were searched against a UniProt *Mus musculus* protein database using TDPortal as previously described (*Fornelli et al., 2017*; *Tyanova et al., 2016*). Annotations were extracted from UniProtKB, Gene Ontology (GO), and the Kyoto Encyclopedia of Genes and Genomes (KEGG). Bioinformatics analyses including hierarchical clustering, pathway analysis, and annotation enrichment were performed with R v4.1.1 using the DEP package (*Zhang et al., 2018*) employing filtering of phosphopeptides identified in less than one-third of samples, variance stabilization for normalization, and maximal reduction of variation between replicates (*Huber et al., 2002*) and imputation. Tests for significance were based on limma, which have a higher sensitivity compared to other algorithms (*van Ooijen et al., 2018*). To achieve a normal distribution, we log2 transformed data after normalization. Missing data, which due to matrix effects are common to proteomics data, were not missing at random; we therefore tested conclusions using the MinProb, man, and QRILC imputation methods appropriate for left censored data (*Lazar et al., 2016*). For all analyses, we used the MinProb algorithm as it yielded the lowest number of significant phosphopeptides while retaining large overlap with the two other methods. Since imputation is non-deterministic, in a minority of cases the IR would not be significantly phosphorylated by insulin. To minimize random effects of imputation, we performed 200 permutations of imputations and testing for significance followed by calculation of the median p-value. Using this algorithm, the IR was consistently found to be phosphorylated by insulin versus BSA, and running the algorithm repeatedly only results in changes to conclusions for 2–3 phosphosites. Pathway enrichment was analyzed using the TopGO package using the classic Kolmogorov–Smirnov test (*Alexa and Rahnenfuhrer, 2010*, topGO: Enrichment Analysis for Gene Ontology. R package version 2.48.0). To calculate the significance between proteomic samples with a sample size of 2, we used the DEP package. The DEP package employs limma for statistical testing. This package uses the *t*-test except that the standard errors have been moderated across genes, that is, shrunk toward a common value, using a simple Bayesian model (*van Ooijen et al., 2018*). The assumptions for limma are that the data are normally distributed and equal variance between replicates. These assumptions were met after normalization and log2 transformation. For the proteomics analysis, we calculated the significance based on imputed values. Zeros were more abundant for phosphopeptides with an overall low intensity across the samples. Therefore, the choice of imputation method was based on data missing not at random (MNAR). We tested three imputation methods and continued with the one resulting in the lowest number of significant phosphopeptides retaining a

relatively large overlap with other imputation methods. In cases where phosphorylation was displayed as zero, the intensity was zero in the raw dataset. We applied semantic similarity analysis between GO terms followed by principal coordinates analysis to cluster and visualize enrichments. Unless otherwise stated, individual comparisons between conditions across phosphopeptides were performed using empirical Bayes statistics followed by adjustment for multiple testing with p-values of *<0.05, **<0.01, and ***<0.001 to be considered significant. The minimum normalized intensity across the dataset was subtracted from each normalized data point, and phosphorylation was calculated as a fraction of the maximum value of all samples for each phosphopeptide. Plotted GO terms have p-values <0.01 calculated using the classic Kolmogorov–Smirnov test. The distribution diagram of shared and unique phosphosites was obtained by retrieving the selected GO pathways for peptides detected in at least one sample between treatments. Specific details on the significance of each test are described in the figure legends. The raw data for this analysis is presented in *Source data 2*.

## Immunohistochemistry and muscle fiber size quantification

Tissues were snap-frozen in liquid nitrogen-cooled isopentane and cross-sectioned at 10 µm (*Brett et al., 2020*). Sections from muscles were fixed using 4% PFA, permeabilized using 0.2% Triton X-100 in PBS, blocked using 1% BSA in PBS, and incubated with anti-laminin antibody (Millipore, clone A5, Cat# 05-206, 1:200) and then with Alexa Fluor secondary antibodies (#A11007, Invitrogen, 1:1000). Nuclei were counterstained with Hoechst (#33342, Thermo Fisher, 1:1000). Images were acquired using a confocal microscope (Leica TCS SP8) at 63×. For hematoxylin and eosin (H&E) staining, slides were stained with hematoxylin for 3 min, washed with water and 95% ethanol, and stained with eosin for 30 min. Sections were then washed with ethanol and xylene, and mounted with mounting medium. The tissue slides were observed with a Nikon 80i upright light microscope using a ×20 objective lens. Digital images were captured with a Nikon Digital Sight DS-Fi1 color camera and NIS-Elements acquisition software. Muscle fiber sizes for the pectoralis, quadriceps, soleus, gastrocnemius, and tibialis muscle tissues were determined by measuring cross-sectional area ($µm^2$) using the Image J (version 1.53e) software. The muscle fibers were manually outlined to obtain their measurement data. Blind scoring by two independent investigators of the muscle tissues was done to unbiasedly collect data for all categories of mice. The quantification of the average muscle fiber area was performed using n = 3 independent muscle tissues from each genotype, with 2–4 photos taken from each muscle tissue. Approximately 60–100 myofibers were quantified per image, and average area values were calculated for each image. Statistically significant differences were determined using two-way ANOVA.

## Indirect calorimetry, food intake, and physiological measurements

Oxygen consumption ($VO_2$), RER, movement, and food intake in 8-to-12-week-old WT and *Ism1*-KO mice were measured using the environment-controlled home-cage CLAMS system (Columbus Instruments, Columbus, OH) at the Stanford Diabetes Research Center. Mice were maintained on a chow diet (Envigo Teklad Global 18% Protein Rodent Diet 2018) and housed at 20–22°C in the cages for 24 hr prior to the recording. Energy expenditure calculations were not normalized for body weight. RER and locomotion were analyzed using ANOVA. All other calorimetry measurements in mice were analyzed using ANCOVA using the CalR version 1.3 without the remove outliers feature (*Mina et al., 2018*). The two-paw grip strength was measured on a grid with a grip meter. The mice were trained to grasp a horizontal bar while being pulled by their tail. The force (expressed in Newton) was recorded by a sensor. Three trials were combined for analysis.

## Femur length measurement

Both left and right femurs from each mouse were carefully dissected, and the length between the distal and proximal ends of the bone was measured with a ruler. The average value of the left and right femurs of each individual mouse was used for data analysis. Statistical significance was calculated using a two-tailed Student's *t*-test.

## Single-cell RNA sequencing of skeletal muscle in mice

Single-cell RNA sequencing data was reanalyzed from a previously published dataset performed on single-cell suspension from murine tibialis anterior skeletal muscles (*Baht et al., 2020*). Briefly, tibialis anterior muscles from three injured mice were pooled as well as three uninjured mice to generate

two samples used for scRNA-Seq. Three thousand single cells from each of the two samples were barcoded and cDNA generated using 10X Genomics Chromium Drop-seq platform and sequencing on Illumina 2500 platform. 10X Genomics Cell Ranger software was used to demultiplex and align reads. Seurat (V4.2) package was used to perform quality control, sample normalization, and clustering for cell type identification. Downstream analyses included gene analysis of the genes of interests associated with cell clusters.

## Expression and purification of recombinant proteins

The Ism1 proteins were generated by transient transfection of mouse Ism1 with C-terminal Myc-6X-his tag DNA plasmids Addgene (#173046) into Expi293F cells. The Expi293F cell line is mycoplasma negative and has been authenticated with STR. Recombinant proteins were produced in mammalian Expi293F cells using large-scale transient DNA transfection and purified using Cobalt columns and buffer exchanged to PBS. Protein purity and integrity were assessed with SDS-PAGE, Superdex200 size-exclusion column and endotoxin assay. Every protein batch produced was tested for bioactivity by measuring the induction of $pAKT^{S473}$ signaling in 3T3-F442A cells as described previously (*Jiang et al., 2021*). All proteins were aliquoted and stored at –80°C and not used for more than three freeze-thaws.

## Culture and differentiation of C2C12 cells

C2C12 cells (#CRL-1772, ATCC) were cultured in DMEM with 10% FBS. The C2C12 cell line is mycoplasma negative and has been authenticated with STR. Cells were passaged every 2 days and were not allowed to reach more than 70% confluency. C2C12 cells were used in the state of myotubes or myoblasts as indicated in figure legends. To differentiate C2C12 cells from myoblasts to myotubes, cells were cultured in differentiation medium (*Risson et al., 2009*; *Sandri et al., 2004*). Cells with passage numbers 6–11 were used for all experiments.

## In vivo and in vitro protein synthesis

In vivo protein synthesis was measured by incorporation of [35S]-methionine into proteins isolated from skeletal muscle in mice. Briefly, mice fasted for 1 hr were i.p. injected with 2.5 µCi/g [35S]-methionine diluted in saline. Then, 2 hr after injection, the quadriceps muscles were removed, weighed, and snap-frozen in liquid nitrogen. The tissues were homogenized using a hand-held homogenizer in RIPA buffer containing protease inhibitor cocktail (Roche) and centrifuged at 4°C to remove cell debris. Protein concentration in the supernatant was determined by BCA assay (Thermo Fisher Scientific, Waltham, MA), and total protein content was calculated by multiplying the protein concentration by the supernatant volume. Proteins were extracted using TCA precipitation and the radioactivity was counted on a scintillation counter. Protein synthesis in C2C12 myotubes was measured by incorporation of [35S]-methionine into proteins using a modified protocol developed for skeletal myotubes (*Hong-Brown et al., 2007*; *Kazi and Lang, 2010*; *Méchin et al., 2007*) as described previously (*Schmidt et al., 2009*). For [35S]-methionine incorporation, C2C12 cells were treated with BSA, recombinant Ism1 or Igf1 with the addition of 0.5 µCi [35S]-methionine (#NEG009L005MC, PerkinElmer, Waltham, MA) for 48 hr. For [35S]-methionine incorporation in the presence or absence of inhibitors, C2C12 cells were treated with DMSO or 100 nM rapamycin for 2 hr, followed by treatments with BSA, recombinant Ism1 or Igf1 for 1 hr. Subsequently, 0.5 µCi [35S]-methionine (#NEG009L005MC, PerkinElmer) was added for another 1 hr. To stop the incubation, cells were washed in ice-cold PBS three times. Proteins were extracted using TCA precipitation, and the radioactivity was counted on a scintillation counter.

## Gene expression analysis

Total RNA from cultured cells or tissues was isolated using TRIzol (Thermo Fisher Scientific) and RNeasy mini kits (QIAGEN, Hilden, Germany). RNA was reverse transcribed using the ABI high-capacity cDNA synthesis kit. For q-RT-pcr analysis, cDNA, primers, and SYBR-green fluorescent dye (Bimake, Houston, TX) were used. Relative mRNA expression was determined by normalization to cyclophilin levels using the ΔΔCt method.

## Western blots and molecular analyses

For Western blotting, homogenized tissues or whole-cell lysates were lysed in RIPA buffer containing protease inhibitor cocktail (Roche, Basel, Switzerland) and phosphatase inhibitor cocktail (Roche), prepared in 4X LDS Sample Buffer (Invitrogen, Waltham, MA) and separated by SDS-PAGE and transferred to Immobilon 0.45 μm membranes (Millipore, Burlington, MA). The cytoplasmic fraction was isolated using a Nuclear Cytoplasmic Extraction Reagent kit (78833, Pierce, Rockford, IL) according to the manufacturer's instructions. The antibodies used are as follows: rabbit monoclonal anti-p-AKT1/2 (Ser473) (#4060), AKT1 (pan) (#4691 CST), rabbit monoclonal anti-p-mTOR (Ser2448) (D9C2), rabbit polyclonal anti-p-S6 ribosomal protein (Ser235/236) (Cat# 2211 CST), and ribosomal protein S6 (#2217 CST) from Cell Signaling. Mouse monoclonal anti-beta actin AC-15 HRP (#AB49900) and OXPHOS rodent antibody (#ab110413) were from Abcam. Donkey anti-rabbit IgG (HRP) (#NA934) and sheep anti-mouse IgG (HRP) (#NA931) were from Cytiva (GE). Recombinant Igf1 (#791-MG-050) was from R&D Systems. The mammalian expression plasmid for Ism1 with C-terminal myc-6xhis tag plasmid for recombinant Ism1 protein production was from Addgene (#173046). The raw Western blot images are presented in *Source data 1*.

## Statistical analyses

Values for *N* represent biological replicates for cultured cell experiments or individual animals for in vivo experiments. Group-housed mice within 8 weeks of age were used for comparative studies. Mice were randomly assigned to treatment groups for in vivo studies. Significant differences between the two groups (*$p<0.05$, **$p<0.01$, ***$p<0.001$) were evaluated using a two-tailed, unpaired Student's *t*-test as the sample groups displayed a normal distribution and comparable variance (Prism9 software; GraphPad). Two-way ANOVA with repeated measures was used for body weight and repeated measurements (*$p<0.05$, **$p<0.01$, ***$p<0.001$). All data are presented as the standard error of mean (SEM) or as described in the figure legends. Specific details for *N* values are noted in each figure legend.

## Acknowledgements

KJS was supported by NIH grants DK125260, DK111916, the Stanford Diabetes Research Center P30 DK116074, the Jacob Churg Foundation, the McCormick and Gabilan Award, the Weintz Family COVID-19 research fund, American Heart Association (AHA), the Stanford School of Medicine, and the Stanford Cardiovascular Institute (CVI). MZ was supported by the American Heart Association (AHA) postdoctoral fellowship (905674). LV was supported by Stanford School of Medicine Dean's Postdoctoral Fellowship. EBM was supported by Stanford School of Medicine Dean's Postdoctoral Fellowship and the American Heart Association (AHA) postdoctoral fellowship (18POST34030448). DEL was supported by NIH training grant T32HL007057. JPW was supported by NIH/NIA grants K01AG056664 and R21AG065943 and Borden Scholar Award through Duke University. NC was supported by the American Heart Association (AHA) SURE award (882082). NBDS was supported by the Carlsberg Foundation Internationalization Fellowship. CVR was supported by the Weintz family COVID-19 research fund, the Stanford Women's Cancer Center and the Stanford Cancer Institute, The Jacob Churg award, and the Children's Cancer Research Foundation. We thank the Stanford University Pathology Histology core facility and the Pathology Department for microscopy equipment. We thank the Proteomics core at Northwestern University for assistance with the phosphoproteomic analysis and Pratima Nallagatla at Stanford University for assistance with the data processing. This work was supported by the Stanford Diabetes Research Center (NIH grant P30DK116074). Some illustrations were created with Biorender under a paid subscription.

# Additional information

## Funding

| Funder | Grant reference number | Author |
| --- | --- | --- |
| National Institute of Diabetes and Digestive and Kidney Diseases | DK125260 | Katrin J Svensson |
| National Institute of Diabetes and Digestive and Kidney Diseases | DK111916 | Katrin J Svensson |
| American Heart Association | 905674 | Meng Zhao |
| American Heart Association | 18POST34030448 | Ewa Bielczyk-Maczynska |
| National Heart, Lung, and Blood Institute | T32HL007057 | David E Lee |
| American Heart Association | 882082 | Nickeisha Cuthbert |
| National Institute of Diabetes and Digestive and Kidney Diseases | DK116074 | Katrin J Svensson |
| National Institute on Aging | R21AG065943 | James P White |
| NIH Office of the Director | K01AG05666 | James P White |

The funders had no role in study design, data collection and interpretation, or the decision to submit the work for publication.

## Author contributions

Meng Zhao, Conceptualization, Data curation, Formal analysis, Funding acquisition, Validation, Investigation, Visualization, Methodology, Writing – original draft, Writing – review and editing; Niels Banhos Danneskiold-Samsøe, Data curation, Formal analysis, Visualization, Writing – original draft, Writing – review and editing; Livia Ulicna, Data curation, Methodology; Quennie Nguyen, Data curation, Investigation; Laetitia Voilquin, Data curation, Investigation, Writing – review and editing; David E Lee, James P White, Data curation, Visualization; Zewen Jiang, Methodology; Nickeisha Cuthbert, Shrika Paramasivam, Data curation; Ewa Bielczyk-Maczynska, Formal analysis, Methodology; Capucine Van Rechem, Funding acquisition, Investigation; Katrin J Svensson, Conceptualization, Resources, Supervision, Funding acquisition, Investigation, Methodology, Writing – original draft, Writing – review and editing

## Author ORCIDs

Meng Zhao ⓘ http://orcid.org/0000-0002-5415-8335
Laetitia Voilquin ⓘ http://orcid.org/0000-0003-2138-4819
Zewen Jiang ⓘ http://orcid.org/0000-0002-3852-8666
Ewa Bielczyk-Maczynska ⓘ http://orcid.org/0000-0002-0558-1188
Capucine Van Rechem ⓘ http://orcid.org/0000-0002-5408-6124
Katrin J Svensson ⓘ http://orcid.org/0000-0001-5376-5128

## Ethics

Animal experiments were performed per procedures approved by the Institutional Animal Care and Use Committee of the Stanford Animal Care and Use Committee (APLAC) protocol number #32982.

## Decision letter and Author response

Decision letter https://doi.org/10.7554/eLife.80014.sa1
Author response https://doi.org/10.7554/eLife.80014.sa2

# Additional files

## Supplementary files
- MDAR checklist
- Source data 1. Unedited raw Western blot images in *Figures 1, 3, 5 and 6*.
- Source data 2. Source data for the phosphoproteomic analysis and quality control.

## Data availability

The phosphoproteomics dataset has been deposited to ProteomeXchange Consortium through JPost PXD031719 (JPST001484) (*Okuda et al., 2017*). The code for all analysis related to phosphoproteomic data is available at https://github.com/Svensson-Lab/Isthmin-1 (copy archived at swh:1:rev:8d60d65f-b98a00d6dfdc8b5fadefe17ce11d25af). The single-cell RNA sequencing data was re-analyzed from a previously published dataset (*Baht et al., 2020*). All the other data generated or analyzed in this study are included in the manuscript and supporting files.

The following dataset was generated:

| Author(s) | Year | Dataset title | Dataset URL | Database and Identifier |
|---|---|---|---|---|
| Zhao M | 2022 | Phosphoproteomic dataset of Ism1 recombinant protein treatment | http://proteomecentral.proteomexchange.org/cgi/GetDataset?ID=PXD031719 | ProteomeXchange, PXD031719 |

The following previously published dataset was used:

| Author(s) | Year | Dataset title | Dataset URL | Database and Identifier |
|---|---|---|---|---|
| Baht GS | 2020 | Single Cell RNAseq on whole muscle one day after injury and uninjured control | https://www.ncbi.nlm.nih.gov/geo/query/acc.cgi?acc=GSE145236 | NCBI Gene Expression Omnibus, GSE145236 |

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

# Appendix 1

## Appendix 1—key resources table

| Reagent type (species) or resource | Designation | Source or reference | Identifiers | Additional information |
|---|---|---|---|---|
| Strain, strain background (*Mus musculus*) | C57BL/6J | Jackson Laboratory | Cat# 000664; RRID:IMSR_ JAX:000664 | |
| Strain, strain background (*M. musculus*) | C57BL/6J Ism1 whole-body knockout | *Jiang et al., 2021* | Cat# 036776 | Generated from C57BL/6J-Ism1em1Kajs/J |
| Cell line (*M. musculus*) | C2C12 | ATCC | CRL-1772; RRID:CVCL_0188 | |
| Cell line (*M. musculus*) | 3T3-F442A | MilliporeSigma | Cat# 00070654; RRID:CVCL_0122 | |
| Cell line (*Homo sapiens*) | Expi293F | Thermo Fisher | Cat#: A14527; RRID:CVCL_D615 | |
| Antibody | Anti-p-AKT (Ser473) (rabbit monoclonal) | Cell Signaling Technology | Cat# 4060; RRID:AB_2315049 | (1:2000) |
| Antibody | Anti-Akt (pan) (rabbit monoclonal) | Cell Signaling Technology | Cat# 4691; RRID:AB_915783 | (1:1000) |
| Antibody | Anti-p-S6 ribosomal protein (Ser235/236) (rabbit polyclonal) | Cell Signaling Technology | Cat# 2211; RRID:AB_331679 | (1:1000) |
| Antibody | Anti-S6 ribosomal protein (rabbit monoclonal) | Cell Signaling Technology | Cat# 2217, RRID:AB_331355 | (1:1000) |
| Antibody | Anti-p-mTOR (Ser2448) (rabbit monoclonal) | Cell Signaling Technology | Cat# 5536; RRID:AB_10691552 | (1:1000) |
| Antibody | Anti-mTOR (rabbit monoclonal) | Cell Signaling Technology | Cat# 2983; RRID:AB_2105622 | (1:1000) |
| Antibody | Anti-beta actin [AC-15] (HRP) (mouse monoclonal) | Abcam | Cat# AB49900; RRID:AB_867494 | (1:25,000) |
| Antibody | Antibody cocktail: OXPHOS Rodent WB Antibody Cocktail (mouse monoclonal) | Thermo Fisher | Cat# 45-8099; RRID:AB_2533835 | (1:1000) |
| Antibody | Anti-phospho-IRS-1 (Ser307) antibody (rabbit polyclonal) | Cell Signaling Technology | Cat# 2381; RRID:AB_330342 | (1:1000) |
| Antibody | Anti-IRS-1 antibody (rabbit monoclonal) | Cell Signaling Technology | Cat# 3407; RRID:AB_2127860 | (1:1000) |
| Antibody | Anti-alpha tubulin antibody (rabbit polyclonal) | Abcam | Cat# ab4074; RRID:AB_2288001 | (1:1000) |
| Antibody | Anti-p27 Kip1 antibody (SX53G8.5) (mouse monoclonal) | Cell Signaling Technology | Cat# 3698; RRID:AB_2077832 | (1:1000) |
| Antibody | IgG HRP linked Ab (rabbit polyclonal) | MilliporeSigma | Cat# NA934; RRID:AB_2722659 | (1:1000) |
| Antibody | IgG HRP linked Ab (mouse monoclonal) | MilliporeSigma | Cat# NA931; RRID:AB_772210 | (1:1000) |
| Antibody | Anti-laminin B2 antibody, clone A5 (rat monoclonal) | MilliporeSigma | Cat# 05-206; RRID:AB_309655 | (1:200) |
| Antibody | Anti-rat IgG secondary antibody, Alexa Fluor 594 (goat polyclonal) | Thermo Fisher Scientific | Cat# A11007; RRID:AB_10561522 | (1:1000) |

*Appendix 1 Continued on next page*

*Appendix 1 Continued*

| Reagent type (species) or resource | Designation | Source or reference | Identifiers | Additional information |
|---|---|---|---|---|
| Sequence-based reagent | Myh1_F | This paper | PCR primer | GCGAATCGAGGCTCAGAACAA |
| Sequence-based reagent | Myh1_R | This paper | PCR primer | GTAGTTCCGCCTTCGGTCTTG |
| Sequence-based reagent | Myh2_F | This paper | PCR primer | AAGTGACTGTGAAAACAGAAGCA |
| Sequence-based reagent | Myh2_R | This paper | PCR primer | GCAGCCATTTGTAAGGGTTGAC |
| Sequence-based reagent | Myh4_F | This paper | PCR primer | TTGAAAAGACGAAGCAGCGAC |
| Sequence-based reagent | Myh4_R | This paper | PCR primer | AGAGAGCGGGACTCCTTCTG |
| Sequence-based reagent | Myh7_F | This paper | PCR primer | ACTGTCAACACTAAGAGGGTCA |
| Sequence-based reagent | Myh7_R | This paper | PCR primer | TTGGATGATTTGATCTTCCAGGG |
| Sequence-based reagent | Ppargc1a_F | This paper | PCR primer | TATGGAGTGACATAGAGTGTGCT |
| Sequence-based reagent | Ppargc1a_R | This paper | PCR primer | CCACTTCAATCCACCCAGAAAG |
| Sequence-based reagent | Foxo1_F | This paper | PCR primer | CCCAGGCCGGAGTTTAACC |
| Sequence-based reagent | Foxo1_R | This paper | PCR primer | GTTGCTCATAAAGTCGGTGCT |
| Sequence-based reagent | Foxo3_F | This paper | PCR primer | CTGGGGGAACCTGTCCTATG |
| Sequence-based reagent | Foxo3_R | This paper | PCR primer | TCATTCTGAACGCGCATGAAG |
| Sequence-based reagent | Foxo4_F | This paper | PCR primer | GGTGCCCTACTTCAAGGACA |
| Sequence-based reagent | Foxo4_R | This paper | PCR primer | AGCTTGCTGCTGCTATCCAT |
| Sequence-based reagent | Cdkn1b_F | This paper | PCR primer | TCAAACGTGAGAGTGTCTAACG |
| Sequence-based reagent | Cdkn1b_R | This paper | PCR primer | CCGGGCCGAAGAGATTTCTG |
| Sequence-based reagent | Eif4ebp1_F | This paper | PCR primer | GGGGACTACAGCACCACTC |
| Sequence-based reagent | Eif4ebp1_R | This paper | PCR primer | CTCATCGCTGGTAGGGCTA |
| Sequence-based reagent | Ctsl_F | This paper | PCR primer | TATCCCTCAGCAAGAGAAAGCCCT |
| Sequence-based reagent | Ctsl_R | This paper | PCR primer | TCCTTCATAGCCATAGCCCACCAA |
| Sequence-based reagent | Fbxo30_F | This paper | PCR primer | TCGTGGAATGGTAATCTTGC |
| Sequence-based reagent | Fbxo30_R | This paper | PCR primer | CCTCCCGTTTCTCTATCACG |
| Sequence-based reagent | UbC_F | This paper | PCR primer | CGTCGAGCCCAGTGTTACCACC |

*Appendix 1 Continued on next page*

*Appendix 1 Continued*

| Reagent type (species) or resource | Designation | Source or reference | Identifiers | Additional information |
|---|---|---|---|---|
| Sequence-based reagent | UbC_R | This paper | PCR primer | ACCTCCCCCATCACACCCAAGA |
| Peptide, recombinant protein | Mouse recombinant Ism1-his protein | This paper and PMID:34348115 | N/A | |
| Peptide, recombinant protein | Recombinant mouse IGF-I/IGF-1 protein | R&D Systems | Cat# 791-MG-050 | |
| Recombinant DNA reagent | Mouse Ism1 with C-terminal Myc-6X-his tag | Addgene and PMID:34348115 | Cat# 173046; RRID:Addgene_173046 | |
| Commercial assay or kit | Pierce BCA Protein Assay Kit | Thermo Fisher Scientific | Cat# 23225 | |
| Commercial assay or kit | Nuclear Cytoplasmic Extraction Reagent kit | Pierce | Cat# 78833 | |
| Chemical compound, drug | Bovine serum albumin | MilliporeSigma | Cat# A7906; CAS:9048-46-8 | |
| Chemical compound, drug | Rapamycin | Cell Signaling Technology | Cat# 9904 | |
| Chemical compound, drug | 2× SYBR Green qPCR master mix | Bimake | Cat# B21203 | |
| Chemical compound, drug | Trizol | Thermo Fisher | Cat# 15-596-026 | |
| Chemical compound, drug | High-capacity cDNA reverse transcription kit | Biosystems | Cat# 4368814 | |
| Chemical compound, drug | DMEM/F12 + GlutaMAX | Thermo Fisher | Cat# 10565-042 | |
| Chemical compound, drug | DMEM high glucose | Sigma | Cat# D6429 | |
| Chemical compound, drug | Trypsin/EDTA 0.25% | Gibco | Cat# 25200-056 | |
| Chemical compound, drug | Penicillin/streptomycin | Gibco | Cat# 15140-122 | |
| Chemical compound, drug | PBS | Gibco | Cat# 10010-023 | |
| Chemical compound, drug | Trypan Blue Stain (0.4%) | Invitrogen | Cat# T10282 | |
| Chemical compound, drug | Hoechst | Thermo Fisher | Cat# 33342 | |
| Chemical compound, drug | Immobilon Crescendo Western HRP substrate | MilliporeSigma | Catt# WBLUR0500 | |
| Chemical compound, drug | SuperSignal West Femto HRP substrate | Thermo Scientific | Cat# 34095 | |
| Chemical compound, drug | SeeBlue Plus2 prestained standard | Invitrogen | Cat# LC5925 | |
| Chemical compound, drug | RIPA buffer (10×) | Cell Signaling | Cat# 9806S | |
| Chemical compound, drug | NuPAGE LDS sample buffer (4×) | Invitrogen | Cat# NP0007 | |
| Chemical compound, drug | 2-Mercaptoethanol | Fisher Chemical | Cat# O3446I | |

*Appendix 1 Continued*

| Reagent type (species) or resource | Designation | Source or reference | Identifiers | Additional information |
|---|---|---|---|---|
| Chemical compound, drug | PhosSTOP | Roche | Cat# 04906837001 | |
| Chemical compound, drug | cOmplete Tablets | Roche | Cat# 04693124001 | |
| Chemical compound, drug | L-[$^{35}$S]-Methionine | PerkinElmer | NEG009L005MC | |
| Software, algorithm | ImageJ | *Schneider et al., 2012* | https://imagej.nih.gov/ij/; RRID:SCR_003070 | |
| Software, algorithm | GraphPad Prism version 8.0 | GraphPad Software, San Diego, CA | http://www.graphpad.com/; RRID:SCR_002798 | |
| Software, algorithm | Adobe Illustrator | Adobe Systems | RRID:SCR_010279 | |
| Software, algorithm | RStudio | https://rstudio.com/ | RRID:SCR_000432 | |

