## [Editor Report]

This article will be of interest to those who study integrated physiology by which muscle size, strength, and metabolism are regulated. Effects of the protein Ism1, which is released by adipocytes and immune cells, on the phosphoproteome were compared and contrasted to those of insulin revealing overlapping though distinct signaling pathways. Ism1 was also shown to determine skeletal muscle size and strength. These data describe a new humoral linkage between fat and skeletal muscle that should have broad implications.

---

## [Decision Letter]

**Decision letter after peer review:**

Thank you for submitting your article "Phosphoproteomic mapping reveals distinct signaling actions and activation of protein synthesis and muscle hypertrophy by Isthmin-1" for consideration by *eLife*. Your article has been reviewed by 3 peer reviewers, including Christopher Cardozo as Reviewing Editor and Reviewer #1, and the evaluation has been overseen by Mone Zaidi as the Senior Editor. The following individual involved in the review of your submission has agreed to reveal their identity: Zachary Graham (Reviewer #2).

Essential revisions:

1) Please rephrase conclusions regarding the effects of the knockout on myofiber size to present a more cautious interpretation of the data.

2) Please carefully consider the remaining comments from each of the three reviewers and revise the manuscript to address them.

*Reviewer #1 (Recommendations for the authors):*

1. A supplemental table showing results ranked using some system should accompany Figure 2P.

2. Figure 1A and corresponding text in legend, results, and methods. It sounds like 6 different samples were pooled, after which the pooled cells were divided into two technical replicates for mass spec. Please revise the text to more clearly reflect what was done.

3. Replotting bar graphs to show individual data points should be performed.

4. Please discuss briefly the fact that all studies were done with male mice as a limitation of the experiments.

5. Line 558 – please indicate how many fibers (approximately) were measured for each muscle.

6. Line 565 – please specify the type of chow – One assumes this was the same feed animals eat normally.

7. Line 566 – please specify how long measurements were recorded.

8. Line 569 – Please specify the type of grip test used: wire, grid, 2-paw, 4-paw. Please indicate the number of trials used and, if appropriate, how data from multiple trials were handled.

9. Line 594 – please indicate the approximate passage number at the time experiments were performed if this is known.

Suggestions:

Line 116 – specify the Akt phosphorylation site in the text for ease of reading.

Including a few literature citations justifying the choice of statistics for the proteomics analysis would improve the manuscript.

*Reviewer #2 (Recommendations for the authors):*

I congratulate the authors for their hard work and innovative study. While there is some enthusiasm for this novel mechanism regulating aspects of muscle health and function, there are some concerns addressed below. I hope the authors do not find the following as negative critiques but rather supportive feedback as the underlying mechanism is quite interesting.

Major limitations

– The varying number of animals per study is kind of confusing, as some activity/metabolic cage data have 4 animals (Figure 4A/B), some molecular work has 3 (Figure 5E) or 5 animals (Figure 5G) grip strength has 15 (Figure 4G), but body weight data only has 3 (Figure 4B). It may just be the way the data are worded and presented. Individual data points on the bar charts would clear this up immediately. In Figure 4 for example, n=3/group but then it is reported that the experiment was repeated using two independent cohorts of mice (line 317). So does n=6? Additionally, methods state animal experiments were sex- and age-matched but only used male mice (line 498). If only male mice were used, what is the justification for excluding female mice from these studies?

– Having only 3 mice per experiment highlights difficulties in interpreting myofiber data. No change in muscle weight weights but myofiber atrophy of only the quad. However, the gastrocnemius fiber size increases by ~20% while the quadriceps has a substantial reduction in fiber size. I don't think the conclusion in line 302 can be interpreted from this data. Ism1 leads to reductions in quad fiber size, yes, but not other muscles. What is unique about the quad for Ism1? Or upper limb function with reduced grip strength?

– Confirming Ism1 doesn't alter canonical IRS phosphorylation sites in your animal model would be very useful. Since Akt S473 is an mTOR target and the cell work shows Ism1-induced protein synthesis is dependent on mTOR, Ism1-KO validation of normal IRS phosphorylation would highlight the distinct nature of Ism1.

Moderate limitations

– While fully understanding the limitations of using cells and the necessity of pooling to get sufficient yield for phosphoproteomics, I have some modest concerns regarding the complete lack of signal in some samples for 2B/E/G. Since phosphorylation state has important biological differences, being completely dephosphorylated in the pooled sample compared to another can have major implications in interpretation. For 2E for example, is TSC2 S1806 completely dephosphorylated by Ism1 or does it have no effect compared to BSA? Regardless of whether this occurred due to the normalization process and is something like a scaling error, some justification for including these and assuming the mean is an appropriate value would be appreciated, even for these early exploratory data.

– Figure 4. Myofiber atrophy is a primary way muscles may lose their mass but also global KO of Ism1 may alter myofiber number, which with the effects of Ism1 on myoblasts, suggests potential developmental roles. Lastly, while the quadriceps/TA/GST is perfectly representative muscle groups to study, they are largely fast-twitch, heavily type IIx/b muscles. Investigations into the soleus or other oxidative muscle would provide information regarding fiber type specificity for mechanisms of action.

– Line 324 Ism1 causes quad myofiber atrophy, not muscle atrophy. And if the marker is myofiber atrophy in the quad, then a 50% reduction in myofiber size as shown in 4F should be considered severe atrophy, not mild.

– Please confirm body mass for 8 wk old animals used for 4B as 30 g would be quite unexpected for an 8 wk old male mouse in my experience.

– No catabolic cell signaling was investigated. While gene markers are understandable and perfectly reasonable, protein level examination of Foxo phosphorylation or other marker associated with muscle atrophy would strongly complement the pAkt/mTOR/S6 data.

*Reviewer #3 (Recommendations for the authors):*

Analysis of Ism1 KO mice revealed a decrease in muscle size (muscle atrophy), but it is unclear whether Ism1 causes muscle hypertrophy. The title and discussion may be overstating somewhat, so I think it is better to reorganize what can be stated from the data.

Since the KO mice used by the author lack Ism1 systemically from birth, whether the decrease in muscle size results from its action on skeletal muscle is unclear. Is there a possibility of a reduction in growth?

Quantification of muscle size is one of the most important data in this study. It was difficult to distinguish the muscle size difference between control and Ism1 KO mice in the photo shown in Figure 4D. The muscle size was quantified using 2-4 photographs taken with a 20× objective lens, but there is concern about whether this is sufficient for accurate measurement since the muscle size differs depending on the region and muscle fiber type. In addition, it is recommended to stain the periphery of cells with laminin or dystrophin to quantify muscle size.

I am concerned that the number of samples is small overall. I have never done phosphoproteomics, but I am unsure whether n = 2 is suitable for statistical analysis.

Line 334-338: Myh 1, 2, 4, and 7 are muscle fiber type markers rather than differentiation markers, so it should be mentioned whether the muscle fiber type has changed.

Line 381: The word "muscle function" is vague, so I think it would be better to rewrite it as a more specific term such as "muscle strength."

Line 381: No data is shown that suggests lsm1 is a "circulating" regulator.

Line 378-38: The same sentence is duplicated.

Line 429: "Ism1-induced muscle hypertrophy" is not shown in the data. To state this, it would be necessary to perform experiments in which Ism1 is administered or overexpressed.

---

## [Author Response]

Essential revisions:1) Please rephrase conclusions regarding the effects of the knockout on myofiber size to present a more cautious interpretation of the data.

We have rephrased the conclusions to describe that Ism1 deficiency causes myofiber atrophy instead of muscle atrophy. We have also added additional sections to discuss the limitations of our study in that only one sex was examined.

2) Please carefully consider the remaining comments from each of the three reviewers and revise the manuscript to address them.

We thank the reviewers for their comments and suggestions. Major changes include adding justification for phosphoproteomic analysis, adding laminin staining for muscle tissues, assays on catabolic cell signaling, and additional data to support that the muscle fiber phenotype was not associated with changes in general growth. Please see the point-by-point responses below to individual reviewers’ comments.

Reviewer #1 (Recommendations for the authors):1. A supplemental table showing results ranked using some system should accompany Figure 2P.

We have added a supplemental table as source data table for Figure 2P (Figure_2_source_data_1) ranked by “Enrichment score”.

2. Figure 1A and corresponding text in legend, results, and methods. It sounds like 6 different samples were pooled, after which the pooled cells were divided into two technical replicates for mass spec. Please revise the text to more clearly reflect what was done.

This is correct. 6 biological replicates per group for the 3 groups (in total 18 samples) were used to reduce cell-to-cell variability. We have now revised the following statements:

Figure legend:

Figure 1A

Line 621 – “(A) Experimental design of the untargeted phosphoproteomics analysis. 3T3-F442A cells were serum-starved for 16 hours and treated with 100 nM recombinant Ism1 or insulin for 5 minutes (*n* = 6 biological replicates per group were pooled and then divided into *n* = 2 technical replicates).”

Results:

Line 118 – “The proteomic experiments were performed in treatment groups of six biological replicates, after which the pooled cells were divided into two technical replicates for the proteomics analysis.”

Methods:

Revised:

Line 403 – “For each treatment group, six biological replicates were pooled into two technical replicates for the phosphoproteomics analysis.”

3. Replotting bar graphs to show individual data points should be performed.

We thank the reviewer for this suggestion. All bar graphs are now revised to show individual data points.

4. Please discuss briefly the fact that all studies were done with male mice as a limitation of the experiments.

Line 390 – In the Methods, “sex- and age-matched, littermate male mice” has been corrected to “age-matched male mice” to clarify that only male mice have been used in this study.

Line 369 – In the Discussion, we have added the limitation “As this study was conducted in male mice only, it is unknown whether Ism1 deficiency results in a muscle phenotype in female mice. Further studies are required to test whether Ism1 regulates muscle proteostasis in a sex-specific manner.”

5. Line 558 – please indicate how many fibers (approximately) were measured for each muscle.

The following sentence has been added to the Methods section:

Line 478 – “The quantification of the average muscle fiber area was performed using *n* = 3 independent muscle tissues from each genotype, with 2-4 photos taken from each muscle tissue. ~60-100 myofibers were quantified per image, and average area values were calculated for each image.”

The following changes have been made to the Figure 4F legend:

Line 693 – “(WT fed, *n* = 3; *Ism1*-KO fed, *n* = 3; WT fasted, *n* = 3; *Ism1*-KO fasted, *n* = 3; one muscle tissue per mouse, 2-4 images per muscle tissue, ~60-100 myofibers quantified per image).”

6. Line 565 – please specify the type of chow – One assumes this was the same feed animals eat normally.

The standard diet information has been added to Methods:

Line 388 – “Envigo Teklad Global 18% Protein Rodent Diet 2018”

7. Line 566 – please specify how long measurements were recorded.

The following sentence has been added to the Results section:

Line 254 – “Ambulatory activity (Figure 5A), respiratory exchange ratio (RER) (Figure 5B), energy expenditure as measured by VO2 consumption (Figure 5C), or food intake (Figure 5D) are indistinguishable between WT and *Ism1*-KO mice when measured over a 48-hour period.”

The following changes have been made to the Figure 5 legend:

Line 706 – “Mice were habituated for 24h followed by 48 h recordings of metabolic parameters”.

8. Line 569 – Please specify the type of grip test used: wire, grid, 2-paw, 4-paw. Please indicate the number of trials used and, if appropriate, how data from multiple trials were handled.

Line 492 – We have edited this section in the Methods to “The 2-paw grip strength was measured on a grid with a grip meter. The mice were trained to grasp a horizontal bar while being pulled by their tail. The force (expressed in Newton) was recorded by a sensor. Three trials were combined for analysis.”

In addition, the following changes have been made to the Figure 4 legend:

Line 697 – “Grip strength measured by 2-paw muscle force (N) on a grid in WT and *Ism1*-KO mice”.

9. Line 594 – please indicate the approximate passage number at the time experiments were performed if this is known.

The following addition has been made to the Methods section:

Line 530 – “Cells with passage numbers 6-11 were used for all experiments.”

Suggestions:Line 116 – specify the Akt phosphorylation site in the text for ease of reading.

The Akt phosphorylation site “S473” has been specified at all places where appropriate.

Including a few literature citations justifying the choice of statistics for the proteomics analysis would improve the manuscript.

We appreciate the reviewer´s comments on statistics which made us reevaluate our data analyses again. We used the variance stabilization normalization (vsn) of the data, as it has been shown to reduce variation most effectively between technical replicates (https://academic.oup.com/bib/article/19/1/1/2562889). Upon reviewer’s request, we reevaluated our normalization. We found that normalization using log2 yielded a more linear qqplot, but did not reduce variation between technical replicates. Normalization using vsn followed by transformation using log2 resulted in a more linear qqplot while retaining low variation between technical replicates. Thus, and log2 transformation following normalization resulted in normally distributed data whereas vsn normalization alone deviated from normality. We therefore reanalyzed our data using normalization followed by log2 transformation of the data. Log2 transformation of the data did not change the conclusions relating to barplots in the main manuscript, except for minor changes in significance levels in Figure 2. Also, the number of significant phosphosites increased slightly from 202 to 221 for differences between BSA and ISM1 (see adjusted supplementary table “Figure_1_source_data_1”). With the exception of Prrc2c^S899^ and Trim56^S450^, all phosphopeptides previously identified as significantly different between BSA and ISM1 were retained.

The documentation for the DEP package suggests filtering the data before testing for significance. We have not been able to find any benchmark for the use of different filters in the literature. Nevertheless, we reviewed the use of a filter for data preprocessing. The filters suggested at the time of data analysis were: (1) keeping peptides identified in all replicates of one condition or (2) keeping peptides identified in two out of three replicates of at least one condition. As we have a sample size of two, as phosphoproteomics is intrinsically sparse, and as the DEP package was made for proteomics and not specifically phosphoproteomics, we considered that filtering would miss valuable information without increasing number of significant peptides. Applying a filter keeping peptides identified in both replicates of one condition only marginally increased the total number of significant proteins. In conjunction with the reevaluation of our data analyses, we revisited the DEP package pipeline and found that it had been updated with less restrictive filters. We tested different filters and found that removing phosphopeptides present in less than 1/3 of samples yielded the highest number of significant differences after adjustment of multiple testing. Comparing BSA to ISM1 we found 297 significant differences in phosphorylation after filtration (see revised supplementary table “Figure_1_source_data_1”).

Next, we reevaluated at the GO terms we included for use in figure. We anticipated that including more goterms including the term “TOR” would yield other phosphosites related to mTOR. We therefore included GO terms including “MTORC1” and “MTORC2” besides “TOR” in the description. This yielded 6 more phosphosites related to mTOR uniquely phosphorylated by ISM (see revised figure 1F) and three new phosphosites with significant differences in phosphorylation between Ism1 and insulin. This included Fnip1^S945^, Rps6^S236 & S240^. We replaced Figure 2E-F with the phosphosites for Rps6 which we find is more in relevant given the increased protein synthesis by Ism1.

In the revised version of the manuscript, we have adjusted the supplementary tables and figures 1E 2A, 2C and 2D to include the added mTOR GO terms. We have changed figures 2I-P using filtering and log2 transformation before imputation and testing. Figure 2E-H were replaced after including additional mTOR GO terms. We have also inserted references in the methods section and changed the text in order to justify the choice of statistics. Importantly, the new analysis results in the same pathways regulated by Ism1.

The following additions have been made to the Methods section for phosphoproteomics analysis:

Line 417 – Data acquired on the Orbitrap were searched against a Uniprot *Mus musculus* protein database using TDPortal as previously described (Fornelli et al., 2017; Tyanova et al., 2016). Annotations were extracted from UniProtKB, Gene Ontology (GO), and the Kyoto Encyclopedia of Genes and Genomes (KEGG). Bioinformatics analyses including hierarchical clustering, pathway analysis and annotation enrichment were performed with R v4.1.1 using the DEP package (Zhang et al., 2018) employing filtering of phosphopeptides identified in less than one third of samples, variance stabilization for normalization and maximal reduction of variation between replicates (Huber et al., 2002) and imputation. Tests for significance were based on limma which have a higher sensitivity compared to other algorithms (van Ooijen et al., 2018). To achieve a normal distribution, we log2 transformed data after normalization. Missing data, which due to matrix effects is common to proteomics data were not missing at random, we therefore tested conclusions using the MinProb, man, and QRILC imputation methods appropriate for left censored data (Lazar et al., 2016). For all analyzes, we used the MinProb algorithm as it yielded the lowest number of significant phosphopeptides while retaining large overlap with the two other methods. Since imputation is non-deterministic, in a minority of cases the insulin receptor would not be significantly phosphorylated by insulin. To minimize random effects of imputation we performed 200 permutations of imputations and testing for significance followed by calculation of the median p-value. Using this algorithm, the insulin receptor was consistently found to be phosphorylated by insulin versus BSA, and running the algorithm repeatedly only result in changed to conclusions for 2-3 phosphosites. Pathway enrichment was analyzed using the TopGO package using the classic Kolmogorov-Smirnov test (Alexa A, Rahnenfuhrer J (2022). topGO: Enrichment Analysis for Gene Ontology. R package version 2.48.0.).

Reviewer #2 (Recommendations for the authors):I congratulate the authors for their hard work and innovative study. While there is some enthusiasm for this novel mechanism regulating aspects of muscle health and function, there are some concerns addressed below. I hope the authors do not find the following as negative critiques but rather supportive feedback as the underlying mechanism is quite interesting.Major limitations– The varying number of animals per study is kind of confusing, as some activity/metabolic cage data have 4 animals (Figure 4A/B), some molecular work has 3 (Figure 5E) or 5 animals (Figure 5G) grip strength has 15 (Figure 4G), but body weight data only has 3 (Figure 4B). It may just be the way the data are worded and presented. Individual data points on the bar charts would clear this up immediately. In Figure 4 for example, n=3/group but then it is reported that the experiment was repeated using two independent cohorts of mice (line 317). So does n=6?

We thank the reviewer for constructive critique and helpful suggestions. The data were collected from several independent cohorts of animals with consistent results, but the cohorts varied in sample size due to differences in litter sizes. In Figure 4, the results were repeated in two independent cohorts (*n* = 3 each cohort, a total of 6 animals), but data from one representative cohort showing identical results using *n* = 3 are presented. To clarify this, we have now revised all bar graphs to show individual data points.

Additionally, methods state animal experiments were sex- and age-matched but only used male mice (line 498). If only male mice were used, what is the justification for excluding female mice from these studies?

In the Methods line 390, “sex- and age-matched, littermate male mice” has been corrected to “age-matched male mice”.

In the Discussion line 369, we have added “As this study was conducted in male mice only, it is unknown whether Ism1 deficiency results in the any muscle phenotype in female mice. Further studies are required to test whether the Ism1 regulates muscle proteostasis in a sex-specific manner.”

– Having only 3 mice per experiment highlights difficulties in interpreting myofiber data. No change in muscle weight weights but myofiber atrophy of only the quad. However, the gastrocnemius fiber size increases by ~20% while the quadriceps has a substantial reduction in fiber size. I don't think the conclusion in line 302 can be interpreted from this data. Ism1 leads to reductions in quad fiber size, yes, but not other muscles. What is unique about the quad for Ism1? Or upper limb function with reduced grip strength?

We agree with the reviewer’s assessment that it is unclear why Ism1 loss only affects fiber size at one of the locations. In the Discussion line 359, we have added “Further studies using larger sample sizes are needed to determine whether the effects of Ism1 on muscle proteostasis are specific to fiber types and locations.”

We have also revised the following sentence in the Results section to only claim what is directly shown.

Line 245 – “…the *Ism1*-KO mice had impaired muscle force (Figure 4G).”

– Confirming Ism1 doesn't alter canonical IRS phosphorylation sites in your animal model would be very useful. Since Akt S473 is an mTOR target and the cell work shows Ism1-induced protein synthesis is dependent on mTOR, Ism1-KO validation of normal IRS phosphorylation would highlight the distinct nature of Ism1.

We have performed additional experiments to show that in our mouse model, IRS phosphorylation at S318 does not differ between WT and *Ism1*-KO mice.

In the Results line 291, we have added “*Ism1*-KO mice are indistinguishable from WT mice in the phosphorylation of IRS1 at S307 (*Figure 6 —figure supplement 1*), a canonical IRS phosphorylation site (Rui et al., 2001).”

In Figure 6 —figure supplement 1, we have added the western blot and quantified protein levels of p-IRS S307 and total IRS1.

Moderate limitations– While fully understanding the limitations of using cells and the necessity of pooling to get sufficient yield for phosphoproteomics, I have some modest concerns regarding the complete lack of signal in some samples for 2B/E/G. Since phosphorylation state has important biological differences, being completely dephosphorylated in the pooled sample compared to another can have major implications in interpretation. For 2E for example, is TSC2 S1806 completely dephosphorylated by Ism1 or does it have no effect compared to BSA? Regardless of whether this occurred due to the normalization process and is something like a scaling error, some justification for including these and assuming the mean is an appropriate value would be appreciated, even for these early exploratory data.

We fully understand the reviewer’s concern. Missing values which are common to proteomics data can either mean that a peptide was not identified or that it was masked by other peptides in the search space. Zeros were more abundant for phosphopeptides with a low intensity. Thus, the true biological value is likely to be low. We calculated significance based on imputed values, and this is what the asterisks in the barplots are based upon. In the cases where phosphorylation was displayed as zero (such as for Figure 2B/E/G) the intensity was zero in the raw dataset. MaxQuant which was used to analyze the phosphoproteomics dataset sets a missing value to zero intensity. Therefore, the normalization in the DEP package interprets zero intensity as ´missing´.

In response to reviewers’ question, we have now better explained the choice of imputation method used for significance testing. We tested algorithms based on the structure of the missing data. Data were not missing at random and left censored (Author response image 1). We sought to be somewhat conservative in our estimate of significant phosphopeptides. We therefore used the imputation algorithm resulting in the lowest number of significant phosphopeptides retaining a relatively large overlap with the other imputation methods (Author response image 2).

**Author response image 1. sa2fig1:** Distribution of missing values as a function of log2 transformed intensities. Missing values (blue) are more abundant among phosphopeptides with a low intensity across samples. This is referred to as left-censored data.

**Author response image 2. sa2fig2:** Overlap between significant phosphopeptides using the missing value imputation algorithms minProb, man and QRILC after running imputation, significance testing and adjustment for multiple testing once for each method. Note that numbers may change slightly from run as imputation algorithms are based upon gaussians.

There is no universally approved method for analyzing missing values. As we do not know the true value of missing points, we chose to set missing values to zero for bar plots, as opposed to an imputed value, or the mean of the average value, which can be misleading. (Please see https://www.rdocumentation.org/packages/imputeLCMD/versions/2.0/topics/impute.MinProb). Nevertheless, we understand that setting the point to zero can be confusing. We have therefore changed the value to n.d. (non-detectable) in bar plots in Figure 2. We have also added that asterisks are based on imputed values. Since we include each measured points and point out missing values in the bar plots, we hope that this is sufficient to clearly interpret the data.

The following additions have been made to the Methods section for phosphoproteomics analysis:

Line 444 – “For the proteomics analysis, we calculated the significance based on imputed values. Zeros were more abundant for phosphopeptides with an overall low intensity across the samples. Therefore, the choice of imputation method was based on data missing not at random (MNAR). We tested three imputation methods and continued with the one resulting in the lowest number of significant phosphopeptides retaining a relatively large overlap with other imputation methods. In cases where phosphorylation was displayed as zero, the intensity was zero in the raw dataset.”

Line 661 – The legend to figure 2 now includes: “Note that, in case of non-detectable phosphopeptides, significance testing was based on imputed values”, and “n.d. = non-detectable”.

– Figure 4. Myofiber atrophy is a primary way muscles may lose their mass but also global KO of Ism1 may alter myofiber number, which with the effects of Ism1 on myoblasts, suggests potential developmental roles. Lastly, while the quadriceps/TA/GST is perfectly representative muscle groups to study, they are largely fast-twitch, heavily type IIx/b muscles. Investigations into the soleus or other oxidative muscle would provide information regarding fiber type specificity for mechanisms of action.

We thank the reviewer for this suggestion. In Figure 4—supplement 2, we have added data from the soleus muscle to show that muscle fiber size and muscle mass were not changed. The figure legend of Figure 4 —figure supplement 2 has been updated accordingly.

Line 359 – In the Discussion, we have added “Further studies using larger sample sizes are needed to determine whether the effects of Ism1 on muscle proteostasis are specific to fiber types and locations.”

– Line 324 Ism1 causes quad myofiber atrophy, not muscle atrophy. And if the marker is myofiber atrophy in the quad, then a 50% reduction in myofiber size as shown in 4F should be considered severe atrophy, not mild.

We have revised the following sentence:

Line 251 – “Since *Ism1* ablation causes severe myofiber atrophy, we next investigated…”

– Please confirm body mass for 8 wk old animals used for 4B as 30 g would be quite unexpected for an 8 wk old male mouse in my experience.

We can confirm that the body mass in our 8-week-old male mice is ~30 g. Jax reports an average body weight of 8-week-old 25.0 ± 1.4 g for wildtype male mice. Our WT and *Ism1*-KO mice are generated from in-house breeding, so the body weight may be different from commercially available mice.

– No catabolic cell signaling was investigated. While gene markers are understandable and perfectly reasonable, protein level examination of Foxo phosphorylation or other marker associated with muscle atrophy would strongly complement the pAkt/mTOR/S6 data.

We have now measured the FoxO target p27 (gene *Cdkn1b*) and observe elevated protein levels in the *Ism1*-KO compared to WT mice, in addition to the previous gene expression data (new Figure 6F). This finding is consistent with increased catabolic signaling in the *Ism1*-KO mice. We have moved the previous Figure 6F on ubiquitin expression into Figure 6 —figure supplement 1.

The Results of this section has been revised to:

Line 279 – Furthermore, under either fed or fasted conditions, muscles isolated from *Ism1*-KO mice have significantly increased transcript levels of atrophy genes FoxO1 (*Figure 6D*), the FoxO target genes *Cdkn1b*, *Eif4ebp1*, and *Ctsl* (*Figure 6E*), ubiquitin ligase *Fbxo30* compared with WT mice (*Figure 6 —figure supplement 1*), and increased protein levels of FoxO target p27 (*Cdkn1b*) (*Figure 6F*) (Zhang et al., 2011).

We have added this to the Methods section Western blots and molecular analyses.:

Line 565 – The cytoplasmic fraction was isolated using a Nuclear Cytoplasmic Extraction Reagent kit (78833, Pierce, Rockford, IL, USA) according to the manufacturer's instructions.

This reference has been added:

Line 1105 – Zhang X, Tang N, Hadden TJ, Rishi AK. Akt, FoxO and regulation of apoptosis. Biochim Biophys Acta. 2011 Nov;1813(11):1978-86. doi: 10.1016/j.bbamcr.2011.03.010. Epub 2011 Mar 31. PMID: 21440011.

Reviewer #3 (Recommendations for the authors):Analysis of Ism1 KO mice revealed a decrease in muscle size (muscle atrophy), but it is unclear whether Ism1 causes muscle hypertrophy. The title and discussion may be overstating somewhat, so I think it is better to reorganize what can be stated from the data.

We agree with the reviewer and have revised the title to:

Line 1 – “Phosphoproteomic mapping reveals distinct signaling actions and activation of muscle protein synthesis by Isthmin-1”

Since the KO mice used by the author lack Ism1 systemically from birth, whether the decrease in muscle size results from its action on skeletal muscle is unclear. Is there a possibility of a reduction in growth?

We thank the reviewer for this insightful comment. We have measured the femur length in male mice at 10 weeks of age and there is no significant difference from WT mice. Therefore, we do not believe that Ism1 deficiency results in growth reduction. We have added Figure 4 – figure supplement 2, methods, results, and discussion to the revised version of the manuscript.

Figure Legend:

Femur length measured in WT and *Ism1*-KO (WT, *n* = 6; *Ism1*-KO, *n* = 5, two-tailed student’s t-test *p < 0.05, **p < 0.01, ***p < 0.001).

Methods:

Line 497 – Femur length measurement

Both left and right femurs from each mouse were carefully dissected, and the length between the distal and proximal ends of the bone was measured with a ruler. The average value of the left and right femurs of each individual mouse was used for data analysis. Statistical significance was calculated using a two-tailed student’s t-test.

Results:

Line 242 – “While the heart weights (*Figure 4 —figure supplement 2*), muscle weights (*Figure 4 —figure supplement 2*), femur length (*Figure 4 —figure supplement 2*), or total body weights (*Figure 4B*) of *Ism1*-KO mice were not significantly different from WT, the *Ism1*-KO mice had impaired muscle force (*Figure 4G*).”

Discussion:

Line 355 – “Femur length is a basic biometric parameter used to assess longitudinal growth. Femur length and total body weights did not differ between WT and *Ism1*-KO mice, suggesting that the decrease in muscle fiber size is likely not due to growth reduction.”

Quantification of muscle size is one of the most important data in this study. It was difficult to distinguish the muscle size difference between control and Ism1 KO mice in the photo shown in Figure 4D. The muscle size was quantified using 2-4 photographs taken with a 20× objective lens, but there is concern about whether this is sufficient for accurate measurement since the muscle size differs depending on the region and muscle fiber type. In addition, it is recommended to stain the periphery of cells with laminin or dystrophin to quantify muscle size.

We apologize for the lack of clarity. The representative pictures shown in figure 4D were taken with a 40x lens to show ~15 myofibers per image. However, the quantification was done using images taken with a 20× objective lens which contain ~ 60-100 myofibers per photo from a total of 2-4 photographs per muscle (*n* = 3 mice/group), totaling to 360-1200 myofibers per condition which we assessed to be sufficient for accurate quantification.

We have magnified the previous H and E staining to show muscle fibers more clearly and moved them to the supplementary figure. We have clarified this in the Methods section and in the Figure legend.

We have also performed laminin staining in the revised Figure 4D to better visualize muscle fibers under fed and fasted conditions in the WT and *Ism1*-KO muscle tissues.

In the Methods section, we have added methods for Immunofluorescence staining and imaging:

Line 464 – “Sections from muscles were fixed using 4% PFA, permeabilized using 0.2% Triton X-100 in PBS, blocked using 1% BSA in PBS, and incubated with anti-Laminin antibody (Millipore, clone A5, cat # 05-206, 1:200) and then with AlexaFluor secondary Antibodies (#A11007, Invitrogen, 1:1000). Nuclei were counterstained with Hoechst (#33342, Thermo Fisher, 1:1000). Images were acquired using a confocal microscope (Leica TCS SP8) at 63x.”

The legends of Figure 4 and Figure 4 —figure supplement 2 have been updated accordingly.

I am concerned that the number of samples is small overall. I have never done phosphoproteomics, but I am unsure whether n = 2 is suitable for statistical analysis.

We thank the reviewer for this important comment and have added additional clarification on how the analysis was done. To calculate the significance, we have used the DEP package that employs limma. We agree that a sample size of two is small. However, there is no lower limit for sample size when using this type of t-test and a sample size of 2 can therefore technically be used. A previous proteomics report indicates that limma accurately estimates significant peptides even at n=2 (van Ooijen et al., 2018). While a larger sample size would be ideal, we do not believe a higher sample size would have changed the results significantly. In many cases signal was completely missing in both replicates of one condition. In such cases, adding another replicate would likely lead to another missing value. Moreover, the number of significant peptides only saturates at a sample size between nine and fifteen. Thus, it is likely that while the number of significant peptides would increase by using a larger sample size, for the significant phosphopeptides identified, our conclusions hold.

The following statement has been added to the Methods section:

Line 438 – “To calculate the significance between proteomic samples with a sample size of 2, we used the DEP package. The DEP package employs limma for statistical testing. This package uses the t-test except that the standard errors have been moderated across genes, i.e., shrunk towards a common value, using a simple Bayesian model (van Ooijen et al., 2018). The assumptions for limma are that the data are normally distributed and equal variance between replicates. These assumptions were met after normalization and log2 transformation“.

The following reference has been added:

Line 1080 – Michiel P van Ooijen, Victor L Jong, Marinus J C Eijkemans, Albert J R Heck, Arno C Andeweg, Nadine A Binai, Henk-Jan van den Ham, Identification of differentially expressed peptides in high-throughput proteomics data, *Briefings in Bioinformatics*, Volume 19, Issue 5, September 2018, Pages 971–981.

Line 334-338: Myh 1, 2, 4, and 7 are muscle fiber type markers rather than differentiation markers, so it should be mentioned whether the muscle fiber type has changed.

We have revised the following sentence in the Results section to reflect the muscle fiber type changes:

Line 261 – “General muscle markers, including the myosin heavy chain proteins *Myh1*, *Myh2*, *Myh4*, and *Myh7* were not altered under fed conditions, and only *Myh4* was reduced in the *Ism1*-KO mice under fasted conditions, suggesting that muscle fiber type is largely unaffected by *Ism1* ablation (*Figure 5H*).”

Line 381: The word "muscle function" is vague, so I think it would be better to rewrite it as a more specific term such as "muscle strength."

We have replaced “muscle function” with “muscle strength” in the following sentence:

Line 293 – “Taken together, these results demonstrate that Ism1 is an anabolic regulator of protein synthesis and muscle strength by maintaining high Akt activity and protein synthesis in skeletal muscle.”

Line 381: No data is shown that suggests lsm1 is a "circulating" regulator.

Ism1 is known to be present in circulation. However, we have removed the word “circulating” on line 294 as we don’t formally show this in this paper in the context of muscle function.

A section in the Discussion section discusses Ism1 as a circulating regulator:

Line 362 – “Circulating levels of human ISM1 using immunoreactive ELISAs have been reported to be in the range of 1-50 ng/ml (Jiang et al., 2021; Ruiz-Ojeda et al., 2022), consistent with mass spectrometry peptide analysis which estimates a circulating Ism1 concentration of 17 ng/ml (Uhlén et al., 2019). The necessary circulating and local Ism1 levels to maintain muscle fiber size or prevent fasting-induced muscle loss remain to be determined.”

Line 378-38: The same sentence is duplicated.

Line 293 – We have removed the duplicated sentence.

Line 429: "Ism1-induced muscle hypertrophy" is not shown in the data. To state this, it would be necessary to perform experiments in which Ism1 is administered or overexpressed.

We have removed the phrase “Ism1-induced muscle hypertrophy” from the following sentence, as we don’t formally show this in this paper:

Line 313 – “Ism1-induced signaling is not simply an activation of the identical insulin/IGF-1-induced PI3K-Akt pathway.”

A section in the Discussion discusses the necessity of future studies to test whether Ism1 induces muscle hypertrophy:

Line 373 – “Long-term studies ought to explore if pharmacological administration of Ism1 systemically or locally into the muscle is sufficient to induce muscle hypertrophy or prevent muscle loss in more severe models of muscle loss.”